# A Unified Domain Adaptation Framework with Distinctive Divergence Analysis

**Zhiri Yuan**                                                              *yuanzhiri2012@gmail.com*
*School of Data Science, City University of Hong Kong*

**Xixu Hu**                                                                 *xixuhu2-c@my.cityu.edu.hk*
*School of Data Science, City University of Hong Kong*

**Qi Wu**                                                                   *qi.wu@cityu.edu.hk*
*School of Data Science, City University of Hong Kong*

**Shumin Ma**                                                              *shuminma@uic.edu.cn*
*Guangdong Provincial Key Laboratory of Interdisciplinary Research and Application for Data Science, BNU-HKBU United International College*

**Cheuk Hang Leung**                                                       *chleung87@cityu.edu.hk*
*School of Data Science, City University of Hong Kong*

**Xin Shen**                                                               *xshen@se.cuhk.edu.hk*
*Department of System Engineering and Engineering Management, The Chinese University of Hong Kong*

**Yiyan Huang**                                                            *yiyhuang3-c@my.cityu.edu.hk*
*School of Data Science, City University of Hong Kong*

**Reviewed on OpenReview:** *https://openreview.net/forum?id=yeT9cBq8Cn*

## Abstract

Unsupervised domain adaptation enables knowledge transfer from a labeled source domain to an unlabeled target domain by aligning the learnt features of both domains. The idea is theoretically supported by the generalization bound analysis in Ben-David et al. (2007), which specifies the applicable task (binary classification) and designates a specific distribution divergence measure. Although most distribution-aligning domain adaptation models seek theoretical grounds from this particular bound analysis, they do not actually fit into the stringent conditions. In this paper, we bridge the long-standing theoretical gap in literature by providing a unified generalization bound. Our analysis can well accommodate the classification/regression tasks and most commonly-used divergence measures, and more importantly, it can theoretically recover a large amount of previous models. In addition, we identify the key difference in the distribution divergence measures underlying the diverse models and commit a comprehensive in-depth comparison of the commonly-used divergence measures. Based on the unified generalization bound, we propose new domain adaptation models that achieve transferability through domain-invariant representations and conduct experiments on real-world datasets that corroborate our theoretical findings. We believe these insights are helpful in guiding the future design of distribution-aligning domain adaptation algorithms.

## 1 Introduction

Domain adaptation (DA) originates from real-world situations where the source and target samples are drawn from different distributions. In many applications, collecting full labels for the target domain is

prohibitively labor-expensive and time-consuming. To correctly label the target domain, DA transfers the knowledge learned from the labeled source domain. Given that distributions of the two domains differ, a remedy to address the distributional difference is to do transfer learning by learning a domain-invariant and task-specific feature space (Ben-David et al., 2010; Tzeng et al., 2014; Ganin et al., 2016; Saito et al., 2018; Xu et al., 2019; Zhang et al., 2019).

To do so, most distribution-aligning DA models base themselves on the theoretical analysis in Ben-David et al. (2007) and use the source error regularized by domain discrepancy as the training objective function. However, the generalization bound given by Ben-David et al. (2007) only applies to the specific binary classification task and the $\mathcal{H}$-divergence that measures the domain similarity. It theoretically restricts the development of DA models for more general multi-class classification and regression tasks (Tzeng et al., 2014; Ganin et al., 2016; Saito et al., 2018; Xu et al., 2019; Sun & Saenko, 2016). Moreover, to measure the domain discrepancy, various distribution divergence measures other than the $\mathcal{H}$-divergence have been applied to DA models (Tzeng et al., 2014; Ganin et al., 2016; Saito et al., 2018; Xu et al., 2019; Long et al., 2017). Thus, a gap exists between the proliferating DA models and the needed theoretical support. In this work, we bridge the theoretical gap by providing a unified generalization bound, regardless of the task type and the divergence measures. Over the various divergence measures, we commit a comprehensive comparison and deliver helpful insights for guiding the future design of distribution-aligning DA algorithms. With our analysis framework, we recover the classical DA models in literature and derive new DA model variants. We empirically show that these model variants can help improve the transfer performance with experiments on classical DA datasets.

We summarize our contributions of this work as follows. (1) We propose a unified generalization bound that can easily adapt to both regression and classification tasks. Meanwhile, it can well accommodate the commonly-used divergence measures, including $f$-divergence, Wasserstein distance, MMD, etc. Thus, it greatly relaxes the stringent constraints imposed by the theoretical foundation in Ben-David et al. (2007). (2) Our analysis underpins and can help recover most classical distribution-aligning DA models in literature. More importantly, new DA models can be derived based on our framework. With experimental studies, we show that these new models greatly enrich the model choices and can even improve the transfer performance. (3) We make a comprehensive comparison of common discrepancies from the perspectives of norm topology, convergence rate and computation cost, which heuristically provides solid theoretical contributions to the DA community.

## 2 Related Work

It is a common belief in DA that minimizing the divergence between source and target domains and the prediction error on the source domain simultaneously could improve the model performance on the target domain. Thus, a large number of previous works focus on constructing suitable regularizing terms to measure the divergence between domains (Redko et al., 2020). Total variation is commonly used to measure the difference between two probability functions, and to our best knowledge it is the first divergence function used in DA (Ben-David et al., 2010). MMD is also popular (Long et al., 2015) in that it can characterize more complicated relations between domains by mapping the input data to a latent feature space. Wasserstein distance is another famous divergence candidate for its neat phisical meaning related to the optimal transport theory (Shen et al., 2018). Recently, many researchers come up with fancier divergences, which may be not natural at first glance but work well in terms of the model performance. For example, Saito et al. (2018) use the task-specific classifiers to construct a regularizer, and Xu et al. (2019) try to align the classifiers of both domains to a fixed scalar. In order to maintain the scale of learned features for regression tasks, Chen et al. (2021) align the domains in terms of principal angles and the representation subspace distance. In general, there are numerous divergence functions for DA, but a further discussion on why they could perform well in DA remains untouched in literature.

The theoretical gaps between the target error analysis and vast numbers of DA models in literature have been partially investigated. Ben-David et al. (2007; 2010) firstly prove that the target error could be bounded by source error and $\mathcal{H}$-divergence or the total variation between domains. Later, Long et al. (2015) prove similar results for MMD. However, the theoretical results in Ben-David et al. (2007; 2010) and Long et al.

(2015) only hold for binary classification tasks. Acuna et al. (2021) fix this gap for $f$-divergences and classification tasks, and Shen et al. (2018) generalize the result for Wasserstein-1 distance and regression tasks. In general, it remains a question whether a specific divergence measure theoretically works in that reducing the corresponding divergence between domains could indeed enhance the model performance on the target domain. In this work, we propose a unified generalization bound analysis framework to make up for the needed theoretical support in previous works.

The comparisons between divergence measures have been rarely investigated. Only a few researchers explain the intuition why a discrepancy measure would outperform another, especially for $f$-divergences (Sason & Verdú, 2016). Nowozin et al. (2016) and Arjovsky et al. (2017) argue that MMD and Wasserstein distance could outperform the total variation, but they only compare them from one single aspect. It is quite natural that one model could outperform or underperform others on different tasks, as we will see in the experiments. In this work, we make a more comprehensive comparison over which divergence measure would be beneficial.

## 3 Preliminaries

Let $\mathcal{X}$ and $\mathcal{Y}$ represent the input and output space, respectively. In unsupervised DA, we are given a *source* domain $\mathcal{D}_s = \{(\mathbf{x}_n^s, y_n^s)\}_{n=1}^{N_s} \subset \mathcal{X} \times \mathcal{Y}$ with $N_s$ labeled samples and a *target* domain $\mathcal{D}_t = \{\mathbf{x}_n^t\}_{n=1}^{N_t} \subset \mathcal{X}$ with $N_t$ unlabeled samples. It is assumed that the two domains share the same input space, but are characterized by different distributions on the input space. In this work, we use $P$ to denote the distribution on the input space of the source domain and $Q$ to denote that of the target domain. Let $h_s, h_t : \mathcal{X} \to \mathcal{Y}$ be the true mapping functions on the source domain $\mathcal{D}_s$ and the target domain $\mathcal{D}_t$ respectively. A hypothesis is a function $h : \mathcal{X} \to \mathcal{Y}$, and we denote the hypothesis set as $\mathcal{H} \subset \{h : \mathcal{X} \to \mathcal{Y}\}$. Let $L : \mathcal{Y} \times \mathcal{Y} \to \mathbb{R}_+$ be the loss function. Then the *source error* of a hypothesis $h$ w.r.t. $h_s$ under distribution $P$ is defined as: $\mathcal{L}_P(h, h_s) := \mathbb{E}_{x \sim P} L(h(x), h_s(x))$. Similarly, we can define the *target error* of a hypothesis $h$, namely, $\mathcal{L}_Q(h, h_t) := \mathbb{E}_{x \sim Q} L(h(x), h_t(x))$. A DA task aims to learn a suitable $h \in \mathcal{H}$ from the labeled source samples as well as the unlabeled target samples such that the target error $\mathcal{L}_Q(h, h_t)$ is small.

Can a hypothesis that achieves the minimal source error generalize well on the target domain? At first glance it sounds impossible in that the distributions characterizing the inputs differ and the mapping functions ($h_s$ and $h_t$) may also differ. With Theorem 1, Ben-David et al. (2007) verify that a hypothesis that can work well on the source domain can indeed generalize well on the target domain given that the two domains are similar. It thus theoretically underpins most unsupervised DA models that are later developed.

**Theorem 1.** *(Ben-David et al., 2007) Let $\mathcal{H}$ be a hypothesis set such that $\mathcal{H} = \{h : \mathcal{X} \to \{0, 1\}\}$ and the loss function be $L(u, v) = |u - v|$. Let $h^*$ be the optimal hypothesis that achieves the minimum joint error on both the source and target domains: $h^* := \operatorname{argmin}_{h \in \mathcal{H}}[\mathcal{L}_P(h, h_s) + \mathcal{L}_Q(h, h_t)]$, and let $\lambda^*$ denote the joint error of the optimal hypothesis $h^*$: $\lambda^* := \mathcal{L}_P(h^*, h_s) + \mathcal{L}_Q(h^*, h_t)$. Then, for any $h \in \mathcal{H}$, we have*

$$\mathcal{L}_Q(h, h_t) \leq \mathcal{L}_P(h, h_s) + d_{\mathcal{H}}(P, Q) + \lambda^*. \tag{1}$$

Here, $d_{\mathcal{H}}(P, Q)$ is the $\mathcal{H}$-**divergence** between distributions $P$ and $Q$, and it measures the similarity of the two domains. For binary classification tasks where $\mathcal{H} = \{h : \mathcal{X} \to \{0, 1\}\}$, it is defined as

$$d_{\mathcal{H}}(P, Q) \triangleq 2 \sup_{h \in \mathcal{H}} |\mathbb{E}_{x \sim P} h(x) - \mathbb{E}_{x \sim Q} h(x)|.$$

Theorem 1 confirms the generalization bound in terms of the source error $\mathcal{L}_P(h, h_s)$ and the discrepancy between domains $d_{\mathcal{H}}(P, Q)$. But two elements have been made fixed: the task (binary classification) and the measure of distributional difference ($\mathcal{H}$-divergence). Real-world DA applications seldom fit strictly into the binary classification framework. To extend to more general settings like multi-class classification or even regression tasks, Mansour et al. (2009) take one step further to establish the generalization bound by relaxing the constraints over the hypothesis set $\mathcal{H}$ with the following Theorem 2.

**Theorem 2.** *(Mansour et al., 2009) Assume that the loss function $L$ is symmetric and obeys the triangle inequality. Define $h_s^* := \operatorname*{argmin}_{h \in \mathcal{H}} \mathcal{L}_P(h, h_s)$, and $h_t^* := \operatorname*{argmin}_{h \in \mathcal{H}} \mathcal{L}_Q(h, h_t)$. For any $h \in \mathcal{H}$, the following inequality holds,*

$$\mathcal{L}_Q(h, h_t) \leq \mathcal{L}_P(h, h_s^*) + \operatorname{dist}_L^{\mathcal{H}}(P, Q) + \mathcal{L}_P(h_s^*, h_t^*) + \mathcal{L}_P(h_s^*, h_s). \tag{2}$$

In Eq. (2), the divergence measure used, namely $\text{dist}_L^{\mathcal{H}}(P, Q)$, is the **discrepancy distance**. The discrepancy distance between two distributions $P, Q$ over $\mathcal{X}$ is defined as

$$\text{dist}_L^{\mathcal{H}}(P, Q) \triangleq \sup_{h,h' \in \mathcal{H}} |\mathcal{L}_P(h, h') - \mathcal{L}_Q(h, h')|.$$

Compared to the $\mathcal{H}$-divergence, discrepancy distance is more general. The discrepancy distance is pretty flexible because its power on measuring the distribution distance can be controlled by the richness of the hypothesis set $\mathcal{H}$ and loss functions $L$.

Although Mansour et al. (2009) remove the constraints over the hypothesis set $\mathcal{H}$ to extend the generalization bound analysis to more realistic applications, we identify by looking into Eq. (2) that a hypothesis $h$ can only generalize well on the target domain when the objective function $\mathcal{L}_P(h, h_s^*) + \text{dist}_L^{\mathcal{H}}(P, Q)$ is minimized. *On one hand*, compared to minimizing the true source error $\mathcal{L}_P(h, h_s)$ in Eq. (1), minimizing $\mathcal{L}_P(h, h_s^*)$ results in a suboptimal hypothesis because $h_s^*$ is only an approximate to the true labeling function $h_s$ and is usually implicit in real-world applications. *On the other hand*, in practice, the discrepancy distance $\text{dist}_L^{\mathcal{H}}(P, Q)$ is hard to implement in the deep neural networks. In all of the DA applications, practitioners mainly choose a specific divergence such as the Jensen-Shannon divergence, Wasserstein distance, MMD distance. etc. There exists a gap between the proliferating DA models that takes various divergence measures and the needed theoretical support to verify the effectiveness of the models. What's more, considering there are multiple divergence measure choices, the conundrum arises: what difference does it make if a practitioner chooses MMD distance rather than Wasserstein distance? How to choose the best divergence measure? In this work, we bridge the afforementioned theoretical gap by providing a unified generalization bound, regardless of the task type and the divergence measures. In addition, we provide the detailed comparison of various divergence measures in DA scenario and deliver helpful insights for guiding future design of DA algorithms.

Table 1: Popular $f$-divergences and corresponding $f$ functions.

| $f$-divergence | $f(t)$ | $f$-divergence | $f(t)$ |
|---|---|---|---|
| Kullback-Leibler (KL) | $t \log t$ | Reverse KL | $-\log t$ |
| Neyman $\chi^2$ | $\frac{1}{t} - 1$ | Pearson $\chi^2$ | $t^2 - 1$ |
| Jensen-Shannon (JS) | $t \log t - (1+t) \log \frac{1+t}{2}$ | Squared Hellinger (SH) | $(\sqrt{t} - 1)^2$ |
| Total Variation (TV) | $\frac{1}{2}|t - 1|$ | | |

### 3.1 Divergence Measures between Distributions

Before proceeding to the detailed analysis of the generalization bound, we briefly introduce the divergence measures that have been commonly used in literature.

**$f$-divergence (Sason & Verdú, 2016).** Let $f : \mathbb{R}_+ \to \mathbb{R}$ be a convex, semi-continuous function satisfying $f(1) = 0$. Let $P, Q$ be two distributions, satisfying that $dP$ is absolutely continuous w.r.t. $dQ$, and both are absolutely continuous w.r.t. a base measure $dx$. The $f$-divergence $D_f$ from $P$ to $Q$ is given by

$$D_f(P, Q) \triangleq \int f(\frac{dP}{dQ}) dQ.$$

Different choices of $f$ have been proposed in literature. For a good review, see Pardo (2018). Table 1 lists the popular choices of the $f$-divergence and their corresponding $f$ functions.

**Wasserstein distance (Villani, 2009).** The Wasserstein distance between distributions $P$ and $Q$ can be interpreted as the optimal (minimal) transportation cost of moving the mass from $P$ into the mass of $Q$. Let $p \in [1, +\infty)$ and $d(\cdot, \cdot)$ be a distance function of $\mathcal{X}$. Let $\Pi(P, Q)$ denote the set of all possible joint distributions whose marginals are $P, Q$ respectively. The Wasserstein-$p$ distance between $P$ and $Q$ is given by

$$W_p(P, Q) \triangleq \inf_{\gamma \in \Pi(P,Q)} (\mathbb{E}_{(x,x') \sim \gamma}[d(x, x')^p])^{\frac{1}{p}}.$$

In real applications, the case $p = 1$ is of particular interest. Thus, we mainly consider the Wasserstein-1 distance ($W_1$) in the following context.

**MMD (Sriperumbudur et al., 2009b).** Given a kernel function $k : \mathcal{X} \times \mathcal{X} \to \mathbb{R}$, let $\phi$ be the feature map such that $k(x, x') = \phi(x) \cdot \phi(x')$, and $\mathcal{H}_k$ be the reproducing kernel Hilbert space (RKHS) associated with the kernel $k$. The MMD distance between distributions $P$ and $Q$ is:

$$\mathrm{MMD}_k(P, Q) \triangleq \|\mathbb{E}_{x \sim P} \phi(x) - \mathbb{E}_{x \sim Q} \phi(x)\|_{\mathcal{H}_k}. \tag{3}$$

## 4 A Unified Generalization Bound

In this section, we propose a unified generalization bound that can both accommodate the general DA tasks (classification/regression) and the commonly-used distribution divergence measures. Based on this unified bound, we provide the theoretical support for the DA models in the literature that lack the necessary theoretical analysis. In addition, with this generalization bound in hand, we are able to propose new DA models which replenish the current DA literature and greatly enrich the model choices. We will empirically verify the efficiency of these new models on benchmark DA datasets in Section 6. In this section, we begin with Theorem 3 where we prove that the target error can actually be upper bounded by the true source error together with the domain difference plus a constant for any hypothesis set and any symmetric loss function that satisfies triangular inequality. The proof of theorems and propositions in the following context is deferred to Appendix A.

**Theorem 3.** *Assume the loss function $L$ is symmetric and satisfies triangular inequality. For any hypothesis set $\mathcal{H}$, it always holds that*

$$\mathcal{L}_Q(h, h_t) \leq \mathcal{L}_P(h, h_s) + \mathrm{dist}_L^{\mathcal{H}}(P, Q) + \lambda^*. \tag{4}$$

At first glance, it seems that Eq. (4) is nothing different compared with Eq. (1), except the divergence term. There are actually improvements in our result. The unified bound in Theorem 3 does not only apply to the binary classification task. We relax the constraints on the loss function and impose no constraints on the hypothesis set, which improve the versatility of such an upper bound for any practical algorithm. Compared to Eq. (2), we move one step further to directly minimize the true source error $\mathcal{L}_P(h, h_s)$ rather than $\mathcal{L}_P(h, h_s^*)$, where estimating $h_s^*$ first can amplify the source error and minimizing the corresponding $\mathcal{L}_P(h, h_s^*)$ will finally produce a suboptimal hypothesis. Hence our proposed Eq. (4) can produce a better hypothesis to generalize well on the target domain.

Although we take the discrepancy distance as the divergence measure as in Eq. (2), we will not end here. We will show the connections between the hypothetical $\mathrm{dist}_L^{\mathcal{H}}(P, Q)$ and the commonly-used divergence measures, including Wasserstein distance, MMD and $f$-divergence. It's already known that Wasserstein-1 distance $(W_1)$ is a special case of discrepancy distance (Shen et al., 2018; Villani, 2009; Sriperumbudur et al., 2009b). In the following analysis, we focus on the other two divergence families and show that the discrepancy distance can actually recover the MMD distance (in Proposition 4) and partially recover the vast $f$-divergence family (in Proposition 6).

**Proposition 4.** *Let $L(u, v) = |u - v|$ and $k$ be a kernel function on $\mathcal{X}$. Let $\mathcal{H}$ consist of functions with norm no greater than 1 in RKHS associated with kernel $k$, then for any two distributions $P$, $Q$, $\mathrm{dist}_L^{\mathcal{H}}(P, Q) = 2\mathrm{MMD}_k(P, Q)$.*

**Lemma 5.** *(Ben-David et al., 2010) Let $L(u, v) = |u - v|$ and $\mathcal{H}$ consists of functions on $\mathcal{X}$ with values in $[0, 1]$, then for any two distributions $P$, $Q$, $\mathrm{dist}_L^{\mathcal{H}}(P, Q) = 2D_{\mathrm{TV}}(P, Q)$.*

**Proposition 6.** *For any $f$-divergence $D_f$ satisfying that $f$ is continuous and strictly convex, there exists a constant $C_f > 0$, such that for any two distributions $P$, $Q$, $D_{\mathrm{TV}}(P, Q) \leq C_f D_f(P, Q)$.*

Proposition 4 verifies that MMD distance can be recovered by the discrepancy distance. With Proposition 6, one can check that the $f$ functions for commonly-used $f$-divergence (including KL, JS, SH, etc.) indeed fit into the continuous and strictly convex conditions. Thus, combining Lemma 5, Proposition 6 with Theorem 3, we see that the inequality in Eq. (4) still holds for if the divergence measure is taken to be these $f$-divergence. It thus confirms that the generalization bound in Theorem 3 can well accommodate the commonly-used $f$-divergence .

Compared with Theorem 1 and 2, Theorem 3 is more general in that it provides theoretical support to most DA algorithms in literature. For example, if we take $L(u, v) = |u - v|$ and let $\mathcal{H}$ consist of all classifiers

takings values in $\{0, 1\}$ (namely, a binary classification task), then the divergence measure $\text{dist}_L^{\mathcal{H}}(P, Q)$ is equal to the $\mathcal{H}$-divergence and we can recover the classical result in Theorem 1. With our approach, we can also theoretically recover the model proposed in Shen et al. (2018), where the authors prove the efficiency of Wasserstein distance measuring the distributional difference for classification tasks. We defer the detailed discussion over the connections with the classical models in the next subsection.

Theorem 3 is also novel in that the underlying analysis framework can overcome the limitations of the generalization bound analysis in previous literature. For example, in Long et al. (2015), the authors prove that using multiple kernel MMD is efficient in classification tasks with the help of Parzen window classifier, which limits the analysis to only classification tasks thus regression tasks do not apply. With our analysis methodology, we can generalize the efficiency analysis to any universal kernel MMD and to both classification and regression tasks. Acuna et al. (2021) prove that $f$-divergence could be used to derive a generalization bound for classifications under a pretty strong assumption on the hypothesis set, while we do not need additional assumptions.

The target error bound analysis that we propose actually still holds irrespective of the problem setting (covariate shift/conditional shift). The difference in solving these two kinds of problems lie in how to guide the algorithm design with the same theoretical target error bound. In the covariate shift setting where it's assumed that the conditional distributions of $Y|X$ between the source and target domains are the same, the third term $\lambda^*$ in the target error bound in Eq. (4) vanishes. Thus, it suffices to just learn the invariant representations while minimizing the source risk, as we have done in the experiments. In general settings where there can be conditional shift, it is not sufficient to simply align the marginal distributions while achieving small error on the source domain, since the third term $\lambda^*$ can not be neglected (as supported by Theorem 3 of [1] and the lower bound on the joint error of source and target domains in Theorem 4.3 of [2]). But we should also note that some researchers point out that the third term can be reduced to a rather small value if the hypothesis space is rich enough ([4]), making it still possible for the model to adapat well to the target domain by only minimizing the domain discrepancy and the source error even if there is conditional distribution shift.

### 4.1 Connections with Classical Models

In Table 2, we list and summarize some classical DA models in literature. From the last column, we see that most models are proposed without the necessary theoretical explanation on why it can generalize well on the target domain. In this section, we take three models (MCD (Saito et al., 2018), AFN (Xu et al., 2019) and DAN (Long et al., 2015)) as examples to illustrate how our theoretical analysis can help explain the underlying mechanism. Our analysis can also recover other benchmark models, which we will omit here.

#### 4.1.1 Model 1: MCD (Saito et al., 2018)

Saito et al. (2018) propose to align source and target features with the task-specific classifiers as a discriminator for a multi-class classification task. Specifically, to incorporate the relationship between class boundaries and target samples, the authors train two discriminators $h_1$ and $h_2$ to maximize the target features' discrepancy. Denote the number of classes by $m$, then the output space $\mathcal{Y} = \{0, 1\}^m$. Let $L(\mathbf{u}, \mathbf{v}) = \frac{1}{m} \sum_{i=1}^m |u_i - v_i|$ where $\mathbf{u}, \mathbf{v} \in \mathbb{R}^m$, then the training objective for the model MCD is

$$\mathcal{L}_P(h_1, h_s) + \mathcal{L}_P(h_2, h_s) - \mathbb{E}_{x \sim Q} L(h_1(x), h_2(x)), \tag{5}$$

Next, we show that minimizing the objective function in Eq. (5) is actually equivalent to minimizing the source error together with measuring the total variation distance between the two domains. Typically, we will prove that $\min -\mathbb{E}_{x \sim Q} L(h_1(x), h_2(x))$ returns the domain divergence $2 * D_{TV}(P, Q)$, given that $h_1$ and $h_2$ can simultaneously perform well on the source domain. Expressing the total variation distance in the form of discrepancy distance, we have

$$D_{\text{TV}}(P, Q) = \frac{1}{2} \sup_{h_1, h_2 \in \mathcal{H}} \left| \mathbb{E}_{x \sim P} L(h_1(x), h_2(x)) - \mathbb{E}_{x \sim Q} L(h_1(x), h_2(x)) \right|.$$

Table 2: Selected models in the literature.

| Literature | Model | Hypothesis set | Divergence measure | Theoretical analysis |
|---|---|---|---|---|
| Ben-David et al. (2007) | – | $\{h : \mathcal{X} \to \{0,1\}\}$ | $\mathcal{H}$-divergence | ✓ |
| Ben-David et al. (2010) | – | $\{h : \mathcal{X} \to \{0,1\}\}$ | Total variation | ✓ |
| Tzeng et al. (2014) | DDC | $\{h : \mathcal{X} \to \{0,1\}^m\}$ | MMD | |
| Long et al. (2015) | DAN | $\{h : \mathcal{X} \to \{0,1\}^m\}$ | MK-MMD | ✓ |
| Sun & Saenko (2016) | CORAL | $\{h : \mathcal{X} \to \{0,1\}^m\}$ | CORAL Loss | |
| Ganin et al. (2016) | DANN | $\{h : \mathcal{X} \to \{0,1\}^m\}$ | Proxy $\mathcal{A}$-distance | |
| Courty et al. (2017) | JDOT | $\{h : \mathcal{X} \to \{0,1\}^m\}$ | Wasserstein distance | ✓ |
| Long et al. (2017) | JAN | $\{h : \mathcal{X} \to \{0,1\}^m\}$ | JMMD | |
| Tzeng et al. (2017) | ADDA | $\{h : \mathcal{X} \to \{0,1\}^m\}$ | – | |
| Long et al. (2018) | CDAN | $\{h : \mathcal{X} \to \{0,1\}^m\}$ | – | |
| Saito et al. (2018) | MCD | $\{h : \mathcal{X} \to \{0,1\}^m\}$ | $L^1$-norm | |
| Sankaranarayanan et al. (2018) | GTA | $\{h : \mathcal{X} \to \{0,1\}^m\}$ | – | |
| Shen et al. (2018) | WDGRL | $\{h : \mathcal{X} \to \{0,1\}^m\}$ | Wasserstein distance | ✓ |
| Xu et al. (2019) | AFN | $\{h : \mathcal{X} \to \{0,1\}^m\}$ | MMFND | |
| Zhang et al. (2019) | MDD | $\{h : \mathcal{X} \to \{0,1\}^m\}$ | Disparity Discrepancy | ✓ |
| Kang et al. (2020) | CAN | $\{h : \mathcal{X} \to \{0,1\}^m\}$ | CDD | |
| Acuna et al. (2021) | $f$-DAL | $\{h : \mathcal{X} \to \{0,1\}^m\}$ | $D_{h,\mathcal{H}}^{\phi}$ discrepancy | ✓ |

By assuming that the classifiers $h_1, h_2$ will work well on the source domain, the authors suggest that the term $\mathbb{E}_{x \sim P} L(h_1(x), h_2(x))$ is small enough to omit. Thus the following equivalence holds:

$$\min_{h_1, h_2} \mathcal{L}_P(h_1, h_s) + \mathcal{L}_P(h_2, h_s) - \mathbb{E}_{x \sim Q} L(h_1(x), h_2(x))$$

$$\Leftrightarrow \min_{h_1, h_2} \mathcal{L}_P(h_1, h_s) + \mathcal{L}_P(h_2, h_s) + \sup_{h_1, h_2} \left| \mathbb{E}_{x \sim Q} L(h_1(x), h_2(x)) \right|$$

$$\Leftrightarrow \min_{h_1, h_2} \mathcal{L}_P(h_1, h_s) + \mathcal{L}_P(h_2, h_s) + \sup_{h_1, h_2} \left| \mathbb{E}_{x \sim P} L(h_1(x), h_2(x)) - \mathbb{E}_{x \sim Q} L(h_1(x), h_2(x)) \right|$$

$$\Leftrightarrow \min_{h_1, h_2} \mathcal{L}_P(h_1, h_s) + \mathcal{L}_P(h_2, h_s) + 2D_{\text{TV}}(P, Q).$$

### 4.1.2   Model 2: AFN (Xu et al., 2019)

Xu et al. (2019) point out that sometimes the hypothesis set $\mathcal{H}$ is so rich that the upper bound will greatly deviate from zero. To restrict $\mathcal{H}$, the authors place a restrictive scalar $R$ to match the corresponding mean feature norms. With a hyperparameter $\lambda \in \mathbb{R}_+$, the objective function to train the model AFN is

$$\mathcal{L}_P(h, h_s) + \lambda \big( \mathcal{L}_P(h, R) + \mathcal{L}_Q(h, R) \big). \tag{6}$$

Now we use our analysis methodology to theoretically prove that Eq. (6) is actually an upper bound of the target error. Notice that

$$\mathcal{L}_Q(h, h_t) \leq \mathcal{L}_Q(R, h_t) + \mathcal{L}_Q(h, R)$$

$$\leq \mathcal{L}_Q(R, h_t) + \mathcal{L}_P(h, R) + \left| \mathcal{L}_P(h, R) - \mathcal{L}_Q(h, R) \right|$$

$$\leq \mathcal{L}_P(h, h_s) + \mathcal{L}_P(h, R) + \mathcal{L}_Q(h, R) + \mathcal{L}_Q(R, h_t) + \mathcal{L}_P(R, h_s).$$

Given that $\mathcal{L}_Q(R, h_t)$ and $\mathcal{L}_P(R, h_s)$ are constants, the above calculations verify that Eq. (6) does work as a generalization bound.

### 4.1.3   Model 3: DAN (Long et al., 2015)

Compared with Tzeng et al. (2014) who firstly use the Gaussian kernel MMD between features of the last layer as the regularizer, the DAN model is novel in that the authors use the multiple-Gaussian kernel function

and consider the sum of MMD for features in multiple layers. Now we show that our analysis framework can incorporate the DAN model.

First, a multiple-Gaussian kernel function is still a kernel function (Berlinet & Thomas-Agnan, 2011), thus Theorem 3 and Proposition 4 guarantee that using the multiple-Gaussian kernel MMD can efficiently bound the target error. By abusing the notations in Long et al. (2015), we denote the feature space after $l$-th layer of the source domain (resp. target domain) by $\mathcal{D}_s^l$ (resp. $\mathcal{D}_t^l$). The objective function of DAN is

$$\mathcal{L}_P(h, h_s) + \lambda \sum_{l=6}^{8} \text{MMD}_k^2(\mathcal{D}_s^l, \mathcal{D}_t^l),$$

where the sum is taken through the squared MMDs between features of the 6th, 7th and 8th layer. Let $\mathcal{D}_s^{\text{sum}} = \bigoplus_{l=6}^{8} \mathcal{D}_s^l$, $\mathcal{D}_t^{\text{sum}} = \bigoplus_{l=6}^{8} \mathcal{D}_t^l$, where $\bigoplus$ is the direct sum, then it can be verified that $\text{MMD}_k^2(\mathcal{D}_s^{\text{sum}}, \mathcal{D}_t^{\text{sum}}) = \sum_{l=6}^{8} \text{MMD}_k^2(\mathcal{D}_s^l, \mathcal{D}_t^l)$. Thus by viewing $\mathcal{D}_s^{\text{sum}}$ and $\mathcal{D}_t^{\text{sum}}$ as the source and target domains, it is reasonable to take $\text{MMD}_k(\mathcal{D}_s^{\text{sum}}, \mathcal{D}_t^{\text{sum}})$ to construct the theoretical upper bound of the target domain, which means that the objective function in Long et al. (2015) can upper-bound the target error.

## 5 Comparison of Divergence Measures

We have shown that the generalization bound in Eq. (4) can admit most of the distribution divergence measures, and such conclusion is verified by the abundant literature in Table 2. A natural question is, is there any difference on choosing another divergence measure? If so, will there be any guidance on how to choose the suitable divergence measure? Through this section, we illustrate in detail the difference of the various divergence measures. In Section 5.3, we conclude that there is a tradeoff in the selection of a specific divergence measure, and thus we leave the second question for open discussion.

### 5.1 Topologies Induced by Metrics

When measuring the difference between distributions $P$ and $Q$, we hope that the discrepancy measure could truly reflect the extent of the difference. Otherwise, it is unclear whether minimizing the domain discrepancy can indeed align the feature spaces. In this respect, total variation distance is unsatisfactory. Take an extreme example. Let $\mathcal{X} = \mathbb{R}$, and $P_\theta = \delta_\theta, \theta \in \mathbb{R}$, namely, the support of $P_\theta$ contains only the point $\{\theta\}$. Then the total variation distance between $P_0$ and $P_\theta$ is given by

$$D_{\text{TV}}(P_0, P_\theta) = \begin{cases} 0, & \theta = 0 \\ 1, & \theta \neq 0 \end{cases}$$

The corresponding neural network model will fail to minimize $D_{\text{TV}}(P_0, P_\theta)$ w.r.t. $\theta$ if $\theta$ is initialized nonzero by backpropagation, making it impossible to align the two domains. However, Wasserstein distance and MMD are known to be able to get rid of this problem (Arjovsky et al., 2017; Li et al., 2017).

To illustrate the underlying mechanism, we look into the norm topologies induced by the metrics. For a space $\mathcal{X}$ with a metric function $d$, the *norm topology* $\tau$ of $(\mathcal{X}, d)$ consists of the sets that could be written as the union of open balls $B(x, r) = \{x' \in \mathcal{X} : d(x, x') < r\}$ with center $x \in \mathcal{X}$ and radius $r \in \mathbb{R}_+$. If $\mathcal{X}$ is known and fixed, we simply omit $\mathcal{X}$ and say that $\tau$ is induced by $d$. We say a topology $\tau_1$ is *coarser* than $\tau_2$, denoted by $\tau_1 \subset \tau_2$, if any open ball in $\tau_1$ is also open in $\tau_2$. If $\tau_1 \subset \tau_2$ and $\tau_2 \subset \tau_1$, then we say $\tau_1$ and $\tau_2$ are *equivalent*. Roughly speaking, the metric with coarser norm topology should perform better. With the following lemma, we could establish the link between the norm topologies and the corresponding metric functions [1], which will facilitate the later analysis.

**Lemma 7.** *Given two metrics $d_1, d_2$ on $\mathcal{X}$, if there exist $a, b \in \mathbb{R}_+$ such that $d_1(x, x') \leq a \times d_2(x, x')^b, \forall x, x' \in \mathcal{X}$, then the norm topology $\tau_1$ of $(\mathcal{X}, d_1)$ is coarser than the norm topology $\tau_2$ of $(\mathcal{X}, d_2)$.*

---

[1]It is worth noting that some members (although not all) of the $f$-divergence are metric functions, such as the total variation and the SH distance. Thus, the analysis over norm topology still applies to these $f$-divergence.

With Lemma 7, we are able to compare the norm topologies induced by the distance measures in Section 3.1. Assume that the underlying space $\mathcal{X}$ is compact with diameter $M < \infty$. By Theorem 6.15 of Villani (2009), it holds that

$$W_1(P, Q) \leq M \times D_{\text{TV}}(P, Q), \tag{7}$$

implying the topology induced by the Wasserstein-1 distance is coarser than the topology induced by the total variation. As for the MMD distance, for a kernel function $k(\cdot, \cdot)$ satisfying that $C := \sup_{x \in \mathcal{X}} k(x, x) < \infty$ (e.g., a Gaussian kernel, or a Laplacian kernel), by Theorem 14 of Sriperumbudur et al. (2009b), we have

$$\text{MMD}_k(P, Q) \leq \sqrt{C} \times D_{\text{TV}}(P, Q). \tag{8}$$

It also suggests a coarser topology induced by the MMD distance than by the total variation. Thus, the conclusions above help explain why in many DA applications, Wasserstein and MMD distance perform the best. But as we will see in the following analysis and experiments, it is not the norm topology that has the final say over the model performance. The implementation details of the domain divergence also significantly affect the model performance.

## 5.2 Convergence Rate

In this part, we compare the various divergence measures from the perspective of the convergence rates w.r.t. the sample size. In practice, the domain discrepancy between distributions $P$ and $Q$ is estimated by the finite samples from $\mathcal{D}_s$ and $\mathcal{D}_t$. Thus, how the empirical distribution divergence $\text{dist}_L^{\mathcal{H}}(\mathcal{D}_s, \mathcal{D}_t)$ converges to the true domain divergence $\text{dist}_L^{\mathcal{H}}(P, Q)$ is vital to the success of the algorithm. Notice that for arbitrary loss function $L$ and hypothesis class $\mathcal{H}$, it holds that

$$|\text{dist}_L^{\mathcal{H}}(\mathcal{D}_s, \mathcal{D}_t) - \text{dist}_L^{\mathcal{H}}(P, Q)| \leq \text{dist}_L^{\mathcal{H}}(\mathcal{D}_s, P) + \text{dist}_L^{\mathcal{H}}(\mathcal{D}_t, Q).$$

Thus, it is enough to analyze the convergence properties of the discrepancy between a typical distribution and its empirical distribution, namely, $\text{dist}_L^{\mathcal{H}}(\mathcal{D}_s, P)$ (or $\text{dist}_L^{\mathcal{H}}(\mathcal{D}_t, Q)$). The faster the empirical distribution converges to its underlying distribution, the faster will the empirical domain difference converge to the true domain divergence.

We introduce the *convergence rate* (CR) to measure the convergence performance. Fix the underlying space $\mathcal{X}$. Given a distribution $P$ and a set of $N$ realizations $\mathcal{D} = \{x_1, ..., x_N\}$, we denote by $P_N = \sum_{i=1}^N \delta_{x_i}/N$ the empirical distribution of $P$. The convergence rate of a probability distance metric $D(\cdot, \cdot)$ is said be $\alpha$, if for any distribution $P$ on $\mathcal{X}$ and $\delta \in (0, 1)$, there exists a constant $C$ such that the following inequality holds with probability at least $1 - \delta$,

$$D(P_N, P) \leq C N^{-\alpha}.$$

For notation simplicity, we denote a distance metric $D(\cdot, \cdot)$ that has convergence rate $\alpha$ as $\text{CR}(D) = \alpha$. Obviously, for a probability metric $D$, the higher the convergence rate $\alpha$, the faster $D(P_N, P)$ converges w.r.t. the sample size $N$. And thus, the corresponding empirical domain difference will converge to the true domain divergence.

In Proposition 8, we list the convergence rates of total variation distance, Wasserstein-1 distance and MMD. We leave the discussion of Proposition 8 to the next subsection (Section 5.3).

**Proposition 8.** *Assume $\mathcal{X}$ is of dimension $d$.*

(1). *For any $f$-divergence $D_f$ that is indeed metric function (e.g., total variation, SH distance, Pearson $\chi^2$ distance, etc.), $\text{CR}(D_f) \leq \frac{1}{2}$. In particular, $\text{CR}(D_{\text{TV}}) = \frac{1}{2}$.*

(2). *If $\mathcal{X}$ is compact, then $\text{CR}(W_1) = \frac{1}{d}$.*

(3). *(Theorem 7 and Corollary 10 in Sriperumbudur et al. (2009a)) If the kernel function $k$ is characteristic and $\sup_{x \in \mathcal{X}} k(x, x) < \infty$, then $\text{CR}(\text{MMD}_k) = \frac{1}{2}$.*

Table 3: Comparison of the divergence measures commonly used in literature.

|  | $f$-divergences | Wasserstein distance | MMD |
|---|---|---|---|
| Computation cost | Low | Low | High |
| Convergence rate | $\leq \frac{1}{2}$ | $\frac{1}{d}$ | $\frac{1}{2}$ |
| Topology | Fine | Coarse | Coarse |

### 5.3 A Summary

In this part, we summarize the properties of commonly-used divergences and make a brief comparison between them.

$f$-**divergence** is a large divergence family that covers various divergence measures. Though many researchers focus on only a few $f$-divergence, such as the total variation and JS divergence, there can be more divergence choices in the applications. The topology determined by a $f$-divergence is rather fine. However, estimating the $f$-divergence with neural networks can be time-consuming and unstable, as the implementation needs the adversarial networks (Zhang et al., 2019; Acuna et al., 2021).

**Wasserstein distance** enjoys a very concrete intuition and a direct connection to the optimal transport theory (Villani, 2009). Compared with $f$-divergence, it has a weaker topology. Given that directly computing the Wasserstein-$p$ distance for arbitrary $p \geq 1$ is challenging, researchers usually adopt the Wasserstein-1 distance. The Wasserstein-1 distance is in some cases limited in that it only measures the difference between the first-order moments of two distributions (Arjovsky et al., 2017).

**MMD** uses the kernel function to measure the domain discrepancy. Similar to the Wasserstein-1 distance, MMD has a weaker topology than $f$-divergence, thus minimizing the domain discrepancy with MMD can indeed align the feature space. However, unlike Wasserstein-1 distance, MMD (e.g. MMD with Gaussian kernel) could measure higher-order moments of two distributions (Li et al., 2017). More importantly, it is easy to implement in neural networks. In applications, instead of fixing a single kernel MMD, researchers have the freedom to combine Gaussian MMDs with multiple bandwidths (Long et al., 2017; 2015; 2018; Li et al., 2017). However, evaluating the MMD distance encounters the computational cost that grows quadratically with sample size (Arjovsky et al., 2017), which limits the scalability of MMD distance.

We summarize the pros and cons of common divergences in Table 3. We observe that there is no consensus on which divergence measure can uniformly outperform its competitors. That may explain why works taking various divergence measures proliferate in the DA literature.

## 6 Experiments

In this section, we enrich the DA models by proposing new model variants that take divergence measures which are not covered in literature before, including total variation (TV), Neyman $\chi^2$, Pearson $\chi^2$, KL, Reverse KL, SH, JS and MMD with Laplacian kernel functions.

Let's first give a brief description over the proposed models. In the unsupervised DA setting, we are given $N_s$ labeled source samples $\{(\mathbf{x}_n^s, y_n^s)\}_{n=1}^{N_s}$, together with $N_t$ unlabeled samples $\{\mathbf{x}_n^t\}_{n=1}^{N_t}$. We'll first train a feature extractor $F(\cdot; \theta_f)$ parameterized by $\theta_f$ to extract the features. We denote the features extracted as $\mathcal{F}^s$ (for source features) and $\mathcal{F}^t$ (for target features). Namely, $\mathcal{F}^s := \{F(\mathbf{x}_n^s; \theta_f)\}_{n=1}^{N_s}$, $\mathcal{F}^t := \{F(\mathbf{x}_n^t; \theta_f)\}_{n=1}^{N_t}$. The feature extractor is followed by a discriminator network $D(\cdot; \theta_d)$ parameterized by $\theta_d$ to minimize the source error $\mathcal{L}(\{\mathbf{x}_n^s, y_n^s\}; \theta_f, \theta_d)$. Here, $\mathcal{L}$ takes cross-entropy loss for a classification problem and mean squared error for a regression problem. Meanwhile, good transferability requires the alignment of domains. Thus, the feature extractor is required to extract the domain-invariant features, guaranteed by minimizing the distance between the features $\text{dist}_L^{\mathcal{H}}(\mathcal{F}^s, \mathcal{F}^t)$. So the general objective for the proposed models is:

$$\min_{\theta_f, \theta_d} \left\{ \mathcal{L}(\{\mathbf{x}_n^s, y_n^s\}; \theta_f, \theta_d) + \lambda \, \text{dist}_L^{\mathcal{H}}(\mathcal{F}^s, \mathcal{F}^t) \right\},$$

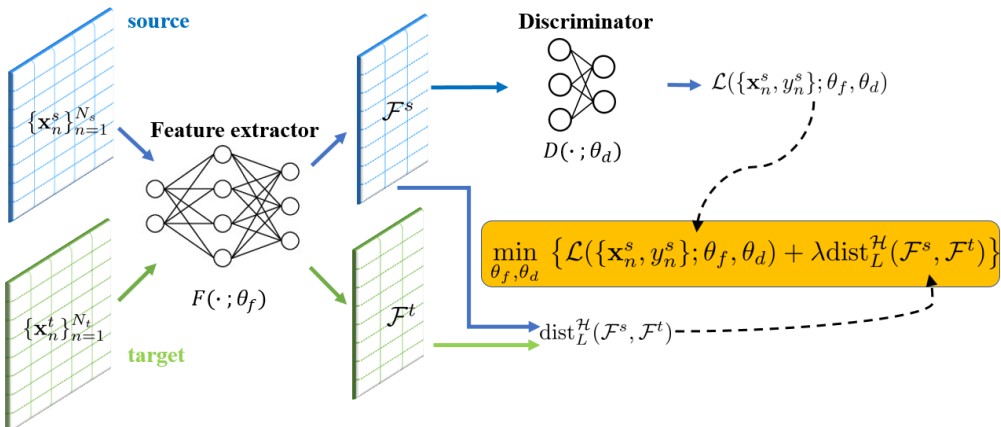

Figure 1: The network flow of the proposed models. All samples, no matter from the source domain $\{\mathbf{x}_n^s\}_{n=1}^{N_s}$ (blue) or the target domain $\{\mathbf{x}_n^s\}_{n=1}^{N_t}$ (green), are fed into a feature extractor $F(\cdot; \theta_f)$ to extract features that are both discriminative and domain-invariant. The discriminative features are achieved by training a discriminator $D(\cdot; \theta_d)$ to minimize the loss function $\mathcal{L}(\cdot; \theta_f, \theta_d)$, while the domain invariance is measured by $\mathrm{dist}_L^{\mathcal{H}}(\mathcal{F}^s, \mathcal{F}^t)$.

where $\lambda$ is a positive hyperparameter to be tuned. For illustration purposes, we sketch the general network flow for the proposed models in Figure 1. For the domain divergence measure $\mathrm{dist}_L^{\mathcal{H}}$ in the proposed models, we take specific divergence measures as mentioned in the last paragraph.

## 6.1 Image Classification

We test the efficacy of the new DA model variants and compare with a variety of baselines on three datasets: **Office-Home**, **Digits** and **Office-31**. All experiments in this section are run on Dell Precision 7920 with Intel® Xeon® Gold 6256 CPU at 3.6GHz, and a set of NVIDIA Quadro GV100 GPU cards.

**Office-Home** consists of 15,500 images in 65 object classes in office and home settings, forming four extremely dissimilar domains: Artistic images (**Ar**), Clipart images (**CA**), Product images (**Pr**), and Real-World images (**Rw**). For the **Digits** dataset, we investigate two digits datasets **MNIST** and **USPS** with two transfer tasks (**M→ U** and **U→M**). We train and evaluate all the methods following the splits and evaluation protocol from Long et al. (2018), which divides **MNIST** into 60,000 training images and 10,000 test images, and divides **USPS** into 7,291 training images and 2,007 test images, respectively. **Office-31** is the most widely used dataset in DA literature. It consists of 4,652 images in 31 categories collected from three sources: Amazon (**A**), Webcam (**W**) and DSLR (**D**). We evaluate all methods on six transfer tasks **A→W**, **D→W**, **W→D**, **A→D**, **D→A**, **W→A**.

**Baseline methods.** We compare with the three models discussed in Section 4.1 and the following classical and state-of-the-art models. JAN (Long et al., 2017) and CAN (Kang et al., 2020) measure the domain discrepancy with variants of MMD. MDD (Zhang et al., 2019) adopts $\gamma$-JSD (a member of $f$-divergence), to align the two domains. $f$−DAL (Acuna et al., 2021) adopts a $f$−divergence variants of MDD to align the two domains. WDGRL (Shen et al., 2018) estimates the domain discrepancy by the Wasserstein distance. DANN (Ganin et al., 2016) analyzes the domain divergence by adversarial neural networks.

**The models we propose.** We propose to use multiple $f$-divergences as regularizers and test the corresponding DA models' effects. Specifically, we use Squared Hellinger distance, Pearson $\chi^2$ distance, KL divergence, total variation, etc. We also test MMD with Laplacian kernel and compare it with the Gaussian kernel. For the Office-Home and Office-31 experiments, ResNet-50 (He et al., 2016) pre-trained on the ImageNet (Deng et al., 2009) is adopted as the backbone feature extractor. For the Digits experiments, LeNet (LeCun et al., 1998) serves as the backbone feature extractor. The classifiers are 2-layer neural networks with width 1024, 2048 and 500 correspondingly. We use separate neural networks to approximate the divergences.

For the Office datasets, we adopt 3-layer neural networks with the same width as the classifiers. For the Digits dataset, we use a 2-layer neural network with width 500.

We follow the standard experiment protocol for unsupervised DA from Ganin et al. (2016) and report the average accuracy for each experiment as in Acuna et al. (2021). All algorithms are implemented in PyTorch. For models using $f$-divergences, we introduce a domain critic inspired by Shen et al. (2018) and Nowozin et al. (2016) to estimate the $f$-divergence in a min-max manner. The domain critic is a 3-layer neural network with ReLU as activation functions. We solve the min-max problem using gradient reversal layers motivated by Ganin et al. (2016). We take the mini-batch SGD optimizer with the Nestorov momentum 0.9 and the learning rates of the classifier are set to be 10 times to that of the feature extractor.

Table 4: Accuracy (%) on the Office-Home datasets in two subtables. The performance with the baseline models are recorded above the middle line and the models we propose below the line in each subtable.

| Method | Ar→Cl | Ar→Pr | Ar→Rw | Cl→Ar | Cl→Pr | Cl→Rw | Avg |
|---|---|---|---|---|---|---|---|
| ResNet-50 | 34.9 | 50.0 | 58.0 | 37.4 | 41.9 | 46.2 | 44.7 |
| DAN(Gaussian) | 46.3 | 66.8 | 74.2 | 57.5 | 63.1 | 66.8 | 62.5 |
| DANN | 45.6 | 59.3 | 70.1 | 47.0 | 58.5 | 60.9 | 56.9 |
| JAN | 45.9 | 61.2 | 68.9 | 50.4 | 59.7 | 61.0 | 57.9 |
| MCD | 53.2 | **73.8** | 77.7 | 62.8 | 68.8 | 70.7 | 67.8 |
| AFN | 52.0 | 71.7 | 76.3 | **64.2** | 69.9 | **71.9** | 67.7 |
| MDD | **54.9** | 73.7 | **77.8** | 60.0 | **71.4** | 71.8 | **68.3** |
| WDGRL | 48.2 | 61.5 | 73.0 | 54.6 | 64.0 | 63.9 | 60.9 |
| CAN | 50.3 | 71.7 | 76.0 | 60.9 | 69.9 | 70.3 | 66.5 |
| $f-$DAL | 53.7 | 71.2 | 76.3 | 60.2 | 68.4 | 69.0 | 66.6 |
| DAN(Laplacian) | 45.7 | 66.5 | 74.5 | 57.0 | 62.4 | 66.5 | 62.1 |
| TV | 46.4 | 65.1 | 73.6 | 55.6 | 64.6 | 66.5 | 62.0 |
| Neyman $\chi^2$ | 50.0 | 66.8 | 75.6 | 56.5 | 66.5 | 67.9 | 63.9 |
| Reverse KL | 55.2 | **74.6** | 78.4 | 57.2 | 69.3 | 71.4 | 67.7 |
| SH | 55.0 | 73.3 | 77.1 | 60.4 | 69.1 | 70.7 | 67.6 |
| JS | 53.5 | 72.9 | 76.5 | 58.8 | 67.3 | 68.6 | 66.3 |
| KL | **55.7** | 73.4 | 77.9 | 60.0 | 70.8 | 72.5 | 68.4 |
| Pearson $\chi^2$ | 55.5 | 69.4 | **78.7** | **61.1** | **73.5** | **72.9** | **68.5** |

| Method | Pr→Ar | Pr→Cl | Pr→Rw | Rw→Ar | Rw→Cl | Rw→Pr | Avg |
|---|---|---|---|---|---|---|---|
| ResNet-50 | 38.5 | 31.2 | 60.4 | 53.9 | 41.2 | 59.9 | 47.5 |
| DAN(Gaussian) | 53.6 | 41.4 | 74.2 | 67.2 | 49.0 | 78.3 | 60.6 |
| DANN | 46.1 | 43.7 | 68.5 | 63.2 | 51.8 | 76.8 | 58.4 |
| JAN | 45.8 | 43.4 | 70.3 | 63.9 | 52.4 | 76.8 | 58.8 |
| MCD | 59.5 | 50.5 | 77.9 | **74.3** | 59.1 | **82.5** | 67.3 |
| AFN | **63.7** | 51.4 | 77.1 | 70.9 | 57.1 | 81.5 | 67.0 |
| MDD | 61.2 | **53.6** | **78.1** | 72.5 | **60.2** | 82.3 | **68.0** |
| WDGRL | 54.3 | 51.2 | 72.9 | 67.9 | 58.9 | 78.1 | 63.9 |
| CAN | 59.0 | 49.7 | 76.2 | 68.5 | 58.5 | 81.8 | 65.6 |
| $f-$DAL | 60.2 | 52.6 | 76.9 | 71.4 | 59.0 | 80.7 | 67.0 |
| DAN(Laplacian) | 53.4 | 40.7 | 73.3 | 66.7 | 48.5 | 78.1 | 60.1 |
| TV | 53.3 | 49.9 | 73.0 | 67.5 | 57.1 | 79.5 | 63.4 |
| Neyman $\chi^2$ | 56.0 | 53.0 | 75.3 | 70.4 | 57.8 | 80.4 | 65.5 |
| Reverse KL | 60.7 | 47.0 | 79.7 | 72.2 | 58.6 | 82.2 | 66.7 |
| SH | 62.5 | 52.9 | 77.7 | 72.3 | 58.0 | 81.4 | 67.5 |
| JS | 61.7 | 52.1 | 76.3 | 71.5 | 57.2 | 80.8 | 66.6 |
| KL | 62.7 | 53.4 | 78.6 | 72.7 | 59.2 | 82.7 | 68.2 |
| Pearson $\chi^2$ | **63.0** | **54.3** | **80.0** | **73.9** | **60.6** | **83.6** | **69.2** |

We record the accuracy results on the Office-Home tasks in Table 4, the results on the Digits tasks in Table 5, and the results on the Office-31 transfer tasks in Table 6. We divide each table with a horizontal line to

Table 5: Accuracy (%) on the Digits datasets.

| Method | M→U | U→M | Avg |
|---|---|---|---|
| DAN (Gaussian) | 88.9 | 87.6 | 88.3 |
| DANN | 91.6 | 95.9 | 93.8 |
| JAN | 84.8 | 90.7 | 87.7 |
| MDD | 93.6 | **97.3** | 95.5 |
| MCD | 94.7 | 96.7 | 95.7 |
| AFN | 89.9 | 96.3 | 93.1 |
| WDGRL | **97.4** | 94.1 | **95.8** |
| CAN | 90.9 | 94.1 | 92.5 |
| $f$−DAL | 91.7 | 95.9 | 93.8 |
| DAN (Laplacian) | 90.1 | 90.1 | 90.1 |
| TV | 91.3 | 96.6 | 94.0 |
| Neyman $\chi^2$ | 92.4 | 95.3 | 93.9 |
| Pearson $\chi^2$ | 92.0 | 95.4 | 93.7 |
| KL | 93.0 | 95.8 | 94.4 |
| Reverse KL | 93.3 | 96.5 | 94.9 |
| SH | **94.8** | **97.3** | **96.1** |
| JS | 93.5 | 96.6 | 95.1 |

Table 6: Accuracy in (%) on the Office-31 datasets.

| Method | A → W | D → W | W → D | A → D | D → A | W → A |
|---|---|---|---|---|---|---|
| ResNet-50 | 68.4 ± 0.2 | 96.7 ± 0.1 | 99.3 ± 0.1 | 68.9 ± 0.2 | 62.5 ± 0.3 | 60.7 ± 0.3 |
| DAN(Gaussian) | 84.3 ± 0.8 | 98.5 ± 0.4 | 100.0 ± 0.0 | 86.1 ± 0.6 | 65.6 ± 0.5 | 64.4 ± 0.4 |
| DANN | 82.0 ± 0.4 | 96.9 ± 0.2 | 99.1 ± 0.1 | 79.7 ± 0.4 | 68.2 ± 0.4 | 67.4 ± 0.5 |
| JAN | 85.4 ± 0.3 | 97.4 ± 0.2 | 99.8 ± 0.2 | 84.7 ± 0.3 | 68.6 ± 0.3 | 70.0 ± 0.4 |
| MCD | 88.6 ± 0.2 | 98.5 ± 0.1 | 100.0 ± 0.0 | 92.2 ± 0.2 | 69.5 ± 0.1 | 69.7 ± 0.3 |
| AFN | 92.5 ± 0.7 | 98.9 ± 0.2 | 100.0 ± 0.0 | **94.8** ± 0.3 | 72.7 ± 0.4 | 70.8 ± 0.6 |
| MDD | 94.5 ± 0.3 | 98.4 ± 0.1 | **100.0** ± 0.0 | 93.5 ± 0.2 | 74.6 ± 0.3 | 72.2 ± 0.1 |
| WDGRL | 92.8 ± 0.3 | 95.6 ± 0.5 | 99.3 ± 0.3 | 88.2 ± 0.2 | 69.9 ± 1.4 | 73.2 ± 0.6 |
| CAN | 93.2 ± 0.2 | 98.4 ± 0.2 | 99.8 ± 0.2 | 92.9 ± 0.2 | **76.5** ± 0.3 | **76.0** ± 0.3 |
| $f$-DAL | **95.4** ± 0.7 | **98.8** ± 0.1 | 100.0 ± 0.0 | 93.8 ± 0.4 | 74.9 ± 1.5 | 74.2 ± 0.5 |
| DAN(Laplacian) | 83.0 ± 0.8 | 98.6 ± 0.3 | 100.0 ± .0 | 85.4 ± 0.6 | 65.6 ± 0.7 | 64.0 ± 0.3 |
| TV | 91.2 ± 1.8 | 98.3 ± 0.1 | 100.0 ± .0 | 89.3 ± 1.0 | 68.1 ± 0.9 | 69.4 ± 1.3 |
| Neyman $\chi^2$ | 88.9 ± 0.9 | 98.0 ± 0.1 | 100.0 ± .0 | 89.8 ± 0.8 | 72.7 ± 1.6 | 72.7 ± 1.3 |
| Pearson $\chi^2$ | **92.5** ± 0.5 | 98.3 ± 0.2 | 100.0 ± .0 | 87.7 ± 0.4 | 72.3 ± 0.6 | **74.4** ± 0.2 |
| KL | 91.1 ± 0.7 | 98.4 ± 0.1 | 100.0 ± .0 | 88.1 ± 0.7 | 75.0 ± 0.00 | 74.3 ± 0.1 |
| Reverse KL | 91.6 ± 0.3 | 98.6 ± 0.1 | 100.0 ± .0 | 88.6 ± 0.8 | 75.1 ± 0.1 | 74.0 ± 0.4 |
| SH | 91.6 ± 0.7 | 98.6 ± 0.1 | 100.0 ± .0 | 89.9 ± 0.8 | 75.0 ± 0.1 | 74.1 ± 0.4 |
| JS | 91.8 ± 1.1 | **98.6** ± 0.1 | **100.0** ± .0 | **90.1** ± 0.4 | **76.1** ± 0.5 | 74.0 ± 0.4 |

separately record the baseline models' performance (above the line with the model name) and our proposed models' performance (below the line with the name of the divergence measure used). We highlight the corresponding highest accuracy among the baseline models and among the models we propose.

From the 3 tables, we make the following observations:

(1) The framework we propose greatly enriches the transfer model choices. Furthermore, some models can even beat the SOTA models. For example, in Table 4, the model with Pearson $\chi^2$ distance achieves the highest average accuracy score among all the DA models. In Table 5, the model using SH distance performs

the best on average. In Table 6, the model using JS divergence achieves the highest accuracy in 4 out of 6 tasks among all the models we propose. Such observations explicitly reveal the great potential of the $f$-divergence family on transfer learning tasks, which fill in the blanks in literature.

(2) But we should also notice that in terms of each specific task in Table 4-6, there does not exist a typical distribution divergence measure that can universally outperform. Besides, for some specific tasks (such as $\mathbf{Ar} \rightarrow \mathbf{Cl}$, $\mathbf{Ar} \rightarrow \mathbf{Pr}$ in Table 4), other models rather than the model using Pearson $\chi^2$ perform better. In addition, there are also tasks in Table 6 where we fail to beat the baseline models. That reminds us to be prudent in selecting a suitable DA model.

(3) Furthermore, DAN with Laplacian kernel MMD performs similarly to DAN with Gaussian kernel MMD, and outperforms the latter one on the Digits dataset in Table 5. It suggests that a wider choice of the kernel functions may be helpful, given that many previous works adopting MMD (Long et al., 2017; 2015; 2018) only consider the Gaussian kernel MMD.

(4) We visualize the features learnt by the models using JS, SH and the other 7 baselines on Office-31 in Figure 2 in the Appendix. It shows that the model with domain similarity measured by JS separates inputs of different classes the best among all. That explains why the model with JS performs pretty well in Table 6.

## 6.2 Image Regression

In addition to the classification tasks, we also test the efficacy of the new DA model variants on two image regression datasets: **MPI3D** and **dSprites**. **MPI3D** is a simulation-to-real dataset of 3D objects, consisting of three domains: Toy (**T**), RealistiC (**RC**) and ReaL (**RL**). Each domain contains 1,036,800 images, with every image having two continuous factors of variations to be predicted: a rotation about a vertical axis at the base and a second rotation about a horizontal axis. We evaluate all methods on six transfer tasks: **RL→RC, RL→T, RC→T, RC→RL, T→RL, and T→RC**. For each task, we separately predict the two rotation angles and record the sum of mean absolute error (MAE) over the two predictions as the transfer performance of this task. **dSprites** is a 2D synthetic dataset composed of three domains: Color (**C**), Noisy (**N**) and Scream (**S**). Each domain consists of 737,280 images with three variants of continuous factors to be predicted: scale, position X, and position Y. We evaluate all methods on 6 transfer tasks: **C→N**, **C→S**, **N→C**, **N→S**, **S→C**, **S→N**. Similar to the experiment on the previous dataset, we record the sum of MAE over the three predictions in each transfer task as the transfer performance. All experiments in this section are run with the same device as in the previous section.

We compare with the following DA models that can be applied to regression problems: DAN, DANN, MCD, AFN and JDOT (Courty et al., 2017). For the MPI3D and dSprites experiments, ResNet-18 (He et al., 2016) pretrained on ImageNet (Deng et al., 2009) is adopted as the backbone feature extractor. The regressors are a 2-layer convolutional neural network with BatchNorm and ReLU as the activation function, followed by an average pooling layer and a 1-layer neural network with width 1024. For models using $f-$divergences, we introduce a domain critic inspired by (Nowozin et al., 2016) to estimate the $f-$divergence in a min-max manner. The domain critic is a 2-layer convolutional neural network followed by a 1-layer neural network with ReLU as activation functions. We solve the min-max problem using gradient reversal layers motivated by (Ganin et al., 2016). We take the mini-batch SGD optimizer with the Nestorov momentum 0.95 and the learning rates of the regressors are set to be 10 times that of the feature extractor.

We follow the standard experimental protocol for unsupervised DA and report the MAE on the two datasets in Table 7 and 8. As in the previous tables, the baseline models' performances are recorded above the line with the model name and our proposed models' performances are below the middle line with the name of the divergence measure used. We highlight the corresponding lowest MAE among the baselines and among the models we propose.

From Table 7 and 8, we can see that the framework we propose enables us to design more domain adaptation regression models with better performance. For example, in Table 7, all the proposed models reduce the MAE by a large margin with Total Variation divergence performing the best. In Table 8, our proposed models also greatly improve the regression performance by lowering the MAE nearly by half. These all demonstrate

the powerful transferability of the proposed model guaranteed by our unified theoretical framework. We should also notice that the performance of different $f-$divergences varies across datasets. For example, the Total Variation divergence performs poorly in the previous two classification tasks, compared with other $f-$divergences like Pearson $\chi^2$. However, the Total Variation divergence achieves the best results in the MPI3D regression tasks. But for the second regression dataset dSprites, the KL divergence and the MMD method with Laplacian kernel achieve the best. These inspire us to try as many as possible divergences in the pilot study period to determine which is the best for our specific task. These could bring significant improvement to the performance. And luckily, our theoretical framework provides a generalization guarantee for a large variety of divergences, which greatly enriches people's choices.

Table 7: MAE on the MPI3D dataset.

| Method | RL→RC | RL→T | RC→RL | RC→T | T→RL | T→RC | Avg |
|---|---|---|---|---|---|---|---|
| ResNet-18 | $0.17 \pm 0.02$ | $0.44 \pm 0.04$ | $0.19 \pm 0.02$ | $0.45 \pm 0.03$ | $0.51 \pm 0.01$ | $0.50 \pm 0.03$ | 0.377 |
| DAN(Gaussian) | $0.12 \pm 0.03$ | $0.35 \pm 0.02$ | $0.12 \pm 0.02$ | $\mathbf{0.27} \pm 0.02$ | $\mathbf{0.40} \pm 0.02$ | $0.41 \pm 0.04$ | **0.278** |
| DANN | $\mathbf{0.09} \pm 0.01$ | $\mathbf{0.24} \pm 0.04$ | $\mathbf{0.11} \pm 0.03$ | $0.41 \pm 0.03$ | $0.48 \pm 0.02$ | $\mathbf{0.37} \pm 0.04$ | 0.283 |
| MCD | $0.13 \pm 0.02$ | $0.40 \pm 0.04$ | $0.15 \pm 0.02$ | $0.45 \pm 0.01$ | $0.52 \pm 0.02$ | $0.50 \pm 0.03$ | 0.358 |
| AFN | $0.18 \pm 0.03$ | $0.45 \pm 0.02$ | $0.20 \pm 0.03$ | $0.46 \pm 0.03$ | $0.53 \pm 0.02$ | $0.52 \pm 0.04$ | 0.390 |
| JDOT | $0.16 \pm 0.02$ | $0.41 \pm 0.01$ | $0.16 \pm 0.02$ | $0.41 \pm 0.02$ | $0.47 \pm 0.02$ | $0.47 \pm 0.02$ | 0.353 |
| DAN(Laplacian) | $0.11 \pm 0.04$ | $0.21 \pm 0.01$ | $0.09 \pm 0.01$ | $0.19 \pm 0.01$ | $0.26 \pm 0.01$ | $0.23 \pm 0.02$ | 0.182 |
| TV | $\mathbf{0.06} \pm 0.00$ | $\mathbf{0.18} \pm 0.01$ | $\mathbf{0.07} \pm 0.02$ | $\mathbf{0.13} \pm 0.02$ | $\mathbf{0.18} \pm 0.01$ | $\mathbf{0.16} \pm 0.01$ | **0.132** |
| Neyman $\chi^2$ | $0.09 \pm 0.02$ | $0.19 \pm 0.00$ | $0.09 \pm 0.01$ | $0.18 \pm 0.01$ | $0.19 \pm 0.00$ | $0.19 \pm 0.00$ | 0.157 |
| Reverse KL | $0.10 \pm 0.00$ | $0.20 \pm 0.00$ | $0.09 \pm 0.00$ | $0.18 \pm 0.01$ | $0.20 \pm 0.01$ | $0.20 \pm 0.00$ | 0.161 |
| SH | $0.11 \pm 0.00$ | $0.21 \pm 0.01$ | $0.10 \pm 0.01$ | $0.20 \pm 0.00$ | $0.21 \pm 0.00$ | $0.21 \pm 0.01$ | 0.176 |
| JS | $0.09 \pm 0.00$ | $0.20 \pm 0.01$ | $0.09 \pm 0.01$ | $0.17 \pm 0.02$ | $0.20 \pm 0.01$ | $0.19 \pm 0.01$ | 0.156 |
| KL | $0.10 \pm 0.01$ | $0.19 \pm 0.00$ | $0.10 \pm 0.01$ | $0.18 \pm 0.01$ | $0.20 \pm 0.00$ | $0.20 \pm 0.01$ | 0.161 |
| Pearson $\chi^2$ | $0.09 \pm 0.02$ | $0.19 \pm 0.00$ | $0.08 \pm 0.00$ | $0.17 \pm 0.01$ | $0.19 \pm 0.00$ | $0.18 \pm 0.00$ | 0.149 |

Table 8: MAE on the dSprites dataset.

| Method | C→N | C→S | N→C | N→S | S→C | S→N | Avg |
|---|---|---|---|---|---|---|---|
| ResNet-18 | $0.94 \pm 0.06$ | $0.90 \pm 0.08$ | $0.16 \pm 0.02$ | $0.65 \pm 0.02$ | $0.08 \pm 0.01$ | $0.26 \pm 0.03$ | 0.498 |
| DAN(Gaussian) | $0.70 \pm 0.05$ | $0.77 \pm 0.09$ | $\mathbf{0.12} \pm 0.03$ | $\mathbf{0.50} \pm 0.05$ | $0.06 \pm 0.02$ | $0.11 \pm 0.04$ | 0.377 |
| DANN | $\mathbf{0.47} \pm 0.07$ | $\mathbf{0.46} \pm 0.07$ | $0.16 \pm 0.02$ | $0.65 \pm 0.05$ | $\mathbf{0.05} \pm 0.00$ | $\mathbf{0.10} \pm 0.01$ | **0.315** |
| MCD | $0.81 \pm 0.09$ | $0.81 \pm 0.12$ | $0.17 \pm 0.12$ | $0.65 \pm 0.03$ | $0.07 \pm 0.02$ | $0.19 \pm 0.04$ | 0.450 |
| AFN | $1.00 \pm 0.04$ | $0.96 \pm 0.05$ | $0.16 \pm 0.03$ | $0.62 \pm 0.04$ | $0.08 \pm 0.01$ | $0.32 \pm 0.06$ | 0.523 |
| JDOT | $0.86 \pm 0.03$ | $0.79 \pm 0.02$ | $0.19 \pm 0.02$ | $0.64 \pm 0.05$ | $0.10 \pm 0.02$ | $0.23 \pm 0.04$ | 0.468 |
| DAN(Laplacian) | $\mathbf{0.18} \pm 0.01$ | $0.28 \pm 0.00$ | $0.08 \pm 0.00$ | $0.23 \pm 0.01$ | $\mathbf{0.05} \pm 0.00$ | $\mathbf{0.07} \pm 0.01$ | **0.151** |
| TV | $0.22 \pm 0.00$ | $0.25 \pm 0.01$ | $0.08 \pm 0.01$ | $0.25 \pm 0.00$ | $0.06 \pm 0.01$ | $0.18 \pm 0.00$ | 0.171 |
| Neyman $\chi^2$ | $0.22 \pm 0.01$ | $0.25 \pm 0.00$ | $0.09 \pm 0.02$ | $0.24 \pm 0.01$ | $0.08 \pm 0.01$ | $0.10 \pm 0.02$ | 0.164 |
| Reverse KL | $0.22 \pm 0.01$ | $0.25 \pm 0.01$ | $0.10 \pm 0.02$ | $\mathbf{0.22} \pm 0.04$ | $0.10 \pm 0.04$ | $0.08 \pm 0.02$ | 0.161 |
| SH | $0.22 \pm 0.00$ | $0.24 \pm 0.02$ | $0.10 \pm 0.02$ | $0.23 \pm 0.02$ | $0.06 \pm 0.00$ | $0.11 \pm 0.01$ | 0.160 |
| JS | $0.22 \pm 0.00$ | $0.24 \pm 0.01$ | $0.08 \pm 0.00$ | $0.23 \pm 0.02$ | $\mathbf{0.05} \pm 0.00$ | $0.11 \pm 0.01$ | 0.155 |
| KL | $0.22 \pm 0.00$ | $\mathbf{0.23} \pm 0.03$ | $\mathbf{0.07} \pm 0.01$ | $0.23 \pm 0.01$ | $0.07 \pm 0.01$ | $0.09 \pm 0.02$ | 0.152 |
| Pearson $\chi^2$ | $0.23 \pm 0.03$ | $0.25 \pm 0.00$ | $0.08 \pm 0.01$ | $0.23 \pm 0.03$ | $0.11 \pm 0.03$ | $0.11 \pm 0.00$ | 0.168 |

# 7 Conclusion

In this work, we focus on the generalization bound for unsupervised domain adaptation (DA) problems. We provide a unified generalization bound to bridge the theoretical gap between the stringent generalization bound analysis in the pioneering work in Ben-David et al. (2007) and the diverse distribution-aligning DA models in literature. This unified bound can well accommodate most commonly-used divergence measures for both classification and regression tasks, and fully recover a large amount of previous models. We further

provide a comprehensive comparison of the commonly-used divergence measures in terms of the norm topology, convergence rate and computation cost. In addition, we propose new DA models that are theoretically guaranteed by the unified generalization bound. We empirically show that a wider range of divergence measures could greatly enrich the model choices and improve the model performance. Our work thus provides theoretical support to many previous models that adopt specific divergence measures and sheds lights on future studies of DA.

## Acknowledgements

We thank the anonymous reviewers from TMLR and our Action Editor for their constructive feedback and thorough suggestions for improvements.

Qi Wu acknowledges the support from the Hong Kong Research Grants Council [General Research Fund 14206117, 11219420, and 11200219], CityU SRG-Fd fund 7005300, and the support from the CityU-JD Digits Laboratory in Financial Technology and Engineering, HK Institute of Data Science. The work described in this paper was partially supported by the InnoHK initiative, The Government of the HKSAR, and the Laboratory for AI-Powered Financial Technologies. Shumin Ma acknowledges the support from: Guangdong Provincial Key Laboratory of Interdisciplinary Research and Application for Data Science, BNU-HKBU United International College (2022B1212010006), Guangdong Higher Education Upgrading Plan (2021-2025) of "Rushing to the Top, Making Up Shortcomings and Strengthening Special Features" with UIC research grant (R0400001-22) and UIC (UICR0700019-22).

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

# A  Appendix

## A.1  Proof

*Proof.* (Theorem 3)

$$
\begin{aligned}
\mathcal{L}_Q(h, h_t) &\leq \mathcal{L}_Q(h, h^*) + \mathcal{L}_Q(h^*, h_t) \\
&\leq \mathcal{L}_P(h, h^*) + |\mathcal{L}_P(h, h^*) - \mathcal{L}_Q(h, h^*)| + \epsilon_T \\
&\leq \mathcal{L}_P(h, h_s) + \mathcal{L}_P(h_s, h^*) + \text{dist}_L^{\mathcal{H}}(P, Q) + \epsilon_T \\
&= \mathcal{L}_P(h, h_s) + \text{dist}_L^{\mathcal{H}}(P, Q) + \epsilon_S + \epsilon_T.
\end{aligned}
$$

$\square$

*Proof.* (Proposition 4) Given the kernel function $k$, by the description of maximum mean discrepancy in Sriperumbudur et al. (2009b), $\text{MMD}_k$ could be equivalently defined as

$$
\text{MMD}_k(P, Q) = \sup_{h \in \mathcal{H}} |\mathbb{E}_{x \sim P}(h(x)) - \mathbb{E}_{x \sim Q}(h(x))|.
$$

Note that $\forall h, h' \in \mathcal{H}$, $h - h'$ is a function in RKHS. Further, we have

$$
\|h - h'\|_{\mathcal{H}} \leq \|h\|_{\mathcal{H}} + \|h'\|_{\mathcal{H}} \leq 2,
$$

namely that $h - h'$ is an element in RKHS with norm no greater than 2. Since there is an explicit bi-jection between $\mathcal{H}$ and the $\mathcal{H}' := \{h \in \text{RKHS} : \|h\|_{\mathcal{H}} \leq 2\}$, given by $h \mapsto 2h$, we have

$$
\begin{aligned}
\text{dist}_L^{\mathcal{H}}(P, Q) &= \sup_{h, h' \in \mathcal{H}} |\mathbb{E}_{x \sim P}|h(x) - h'(x)| - \mathbb{E}_{x \sim Q}|h(x) - h'(x)|| \\
&= 2 \times \sup_{h \in \mathcal{H}} |\mathbb{E}_{x \sim P} h(x) - \mathbb{E}_{x \sim Q} h(x)| \\
&= 2\text{MMD}_k(P, Q).
\end{aligned}
$$

Thus we could conclude that in this case $\text{dist}_L^{\mathcal{H}}$ is two times the $\text{MMD}_k$. $\square$

*Proof.* (Proposition 6) We will refer to the proof of Theorem 3.1 in Csiszár (1972). Let $m = 1, w_m = w_0 = 1, \Pi = \{Q\}$, then in this case, $Q^* = Q$. Let $\delta = \frac{1}{2}, K = 2$, and let $\tilde{f}(t) = tf(\frac{1}{t})$, which is also continuous for $t > 0$ and almost everywhere differentiable. Let $\hat{g}(t) = \tilde{f}'(t)$ at each point $t$ where $\tilde{f}(t)$ is differentiable. Then,

$$\alpha_\delta(u_0) = \min\{\tilde{f}(u_0 + \delta) - \tilde{f}(u_0) - \delta\hat{g}(u_0), \tilde{f}(u_0) - \tilde{f}(u_0 - \delta) - \delta\hat{g}(u_0)\} > 0$$

for $u_0 \in [\delta, K]$ is continuous, and $\alpha_\delta$ is positive and bounded in $[\delta, K]$. Thus

$$\epsilon_{\delta,K} = \frac{1}{\delta} \min_{\delta \leq u_0 \leq K} \alpha_\delta(u_0) > 0$$

is a positive finite constant depending on $f$. Here, we emphasize that the above constants $w_0, \delta, \epsilon_{\delta,K}$ are independent of $Q$. Then for any distribution $P$, by Equation (3.25) of Csiszár (1972),

$$D_{TV}(P, Q) \leq \frac{1}{\epsilon_{\delta,K}(1 - \delta)w_0} D_f(P, Q).$$

$\square$

*Proof.* (Lemma 7) Given any open ball

$$B = B(x_0, r), x_0 \in \mathcal{X}, r \in \mathbb{R}_+$$

in topology $\tau_1$, in order to show that $B$ is also open in $\tau_2$, we need to show that for any $x_1 \in B$, there is an open ball in $\tau_2$ that is inside $B$ and contains $x_1$. Indeed, for any $x_1 \in B$, denote $\delta = d_1(x_0, x_1)$, then $\delta < r$. Consider the open ball $B' = \{x \in X : d_2(x_1, x) < (\frac{r-\delta}{a})^{\frac{1}{b}}\}$. It is direct that for any $z \in B'$, we have

$$d_1(x_0, z) \leq d_1(x_0, x_1) + d_1(x_1, z)$$
$$\leq \delta + ad_2(x_1, z)^b$$
$$< \delta + a \times \frac{r - \delta}{a} = r.$$

Thus $B' \subset B$, hence $B$ is open in $\tau_2$. This shows that $\tau_1 \subset \tau_2$. $\square$

*Proof.* (Proposition 8) Fix an integer $N$. Note that when $\text{dist}_L^{\mathcal{H}}$ takes the form of the total variation or Wasserstein-1 distance, $\text{dist}_L^{\mathcal{H}}$ is bounded. Thus $\text{dist}_L^{\mathcal{H}}(P_N, P)$ as a function of $x_1, ..., x_N$ satisfies the bounded difference property. Indeed, for any $(x_1, ..., x_N), (x_1', ..., x_N') \in \mathcal{X}^N$ that only differs in the $i$-th coordinate (namely, $x_j = x_j'$ for all $j \neq i$), we denote their empirical distributions by $P_N, P_N'$ respectively, there exists $M > 0$ such that

$$|\text{dist}_L^{\mathcal{H}}(P_N, P) - \text{dist}_L^{\mathcal{H}}(P_N', P)| \leq \text{dist}_L^{\mathcal{H}}(P_N, P_N') \leq \frac{M}{N}.$$

By McDiarmid's inequality, with probability at least $1 - \delta$, we shall have

$$\text{dist}_L^{\mathcal{H}}(P_N, P) \leq \mathbb{E}_{P_N}\text{dist}_L^{\mathcal{H}}(P_N, P) + M\sqrt{\frac{1}{2N}\log\frac{1}{\delta}}.$$

Thus we only need to figure out that how large $\mathbb{E}_{P_N}\text{dist}_L^{\mathcal{H}}(P_N, P)$ is. If there exists a constant $\beta$ and a constant $C$ such that

$$\mathbb{E}_{P_N}\text{dist}_L^{\mathcal{H}}(P_N, P) \leq CN^{-\beta},$$

then the convergence rate will be $\min\{\frac{1}{2}, \beta\}$. For the Wasserstein-1 distance, by Theorem 1 in Fournier & Guillin (2015), there exists a constant $C$ such that

$$\mathbb{E}_{P_N}W_1(P_N, P) \leq CN^{-\frac{1}{d}}.$$

For the total variation distance, by Theorem 1 of Rubenstein et al. (2019), there exists a constant $C$ such that

$$\mathbb{E}_{P_N}D_{\text{TV}}(P_N, P) \leq CN^{-\frac{1}{2}}.$$

Furthermore, by Proposition 6, for any $f$-divergence metric $D_f$ satisfying the assumptions, we shall have

$$D_{TV}(P,Q) \leq C_f D_f(P,Q).$$

Thus $\mathbb{E}_{P_N} D_f(P_N, P)$ must converge to 0 with rate $O(N^{-\alpha})$ with $\alpha \leq \frac{1}{2}$. Otherwise, $C_f D_f(P_N, P)$ would be smaller than $D_{TV}(P_N, P)$ when $N$ is sufficiently large, which contradicts Proposition 6. Hence for such metric $D_f$, its convergence rate is no greater than $\frac{1}{2}$.

$\square$

### A.2 Supplementary figures

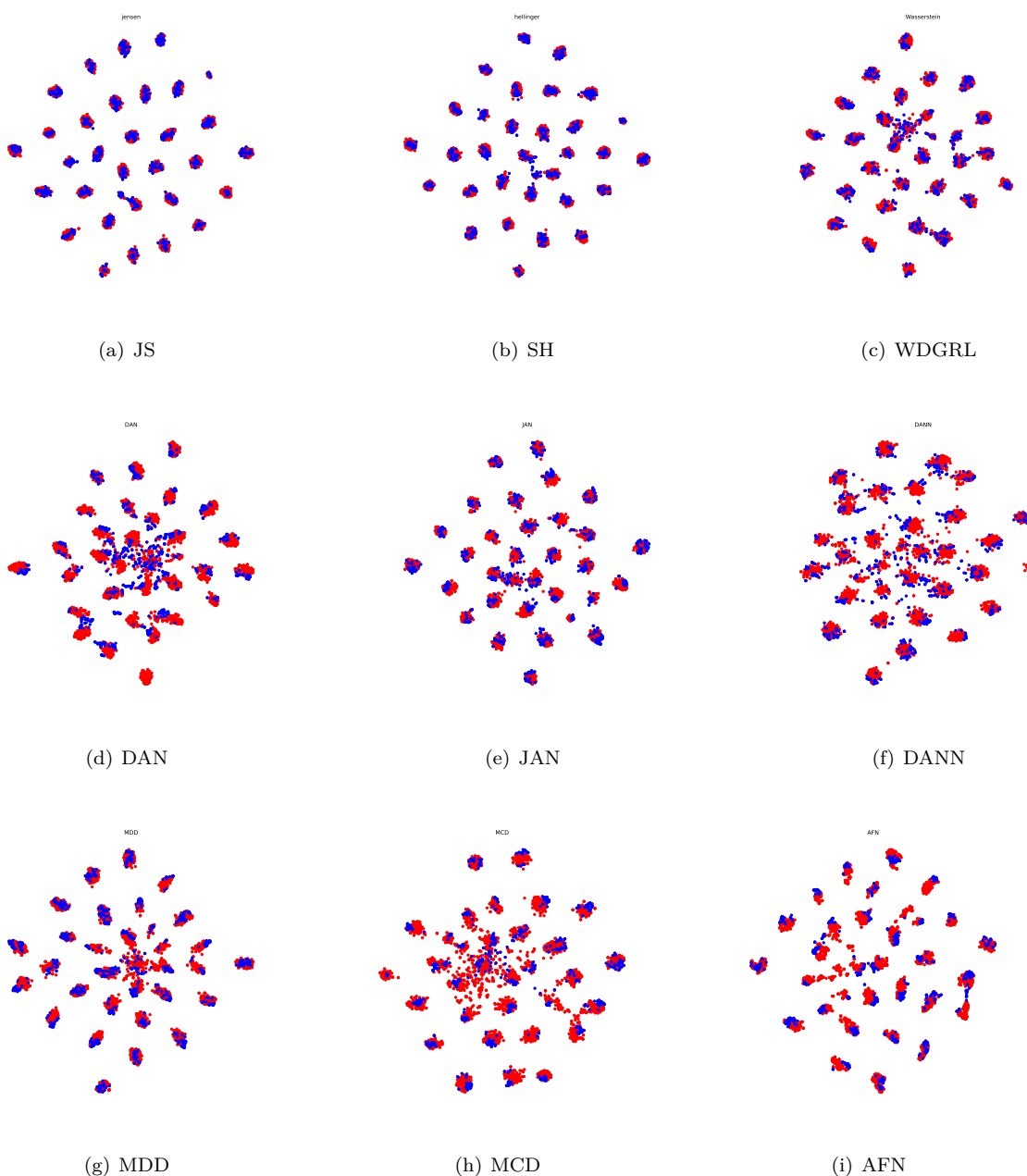

(a) JS  (b) SH  (c) WDGRL

(d) DAN  (e) JAN  (f) DANN

(g) MDD  (h) MCD  (i) AFN

Figure 2: t-SNE visualizations of features learnt from the 9 models on Office-31. The red (blue) points represent the source (target) domain features and one clustering represents one class.

