# OpenReview forum: "A Unified Domain Adaptation Framework with Distinctive Divergence Analysis"
_TMLR — Accepted by TMLR_

### Review · Reviewer_7NW6 · 2022-09-29

**Summary Of Contributions:**

It seems that this paper is a small survey related to the learning bounds and related theoretical applications of domain adaptation.

This paper compares different learning bounds with different distribution discrepancies.

The authors claim that they propose new domain adaptation models and conduct experiments on real-world datasets that corroborate our theoretical findings.

**Broader Impact Concerns:**

No ethical implications

**Requested Changes:**

The paper should discuss  UDA theory when the conditional distributions are different. See weakness 3).





**Strengths And Weaknesses:**

Strength:
1. Well written;
2. A detailed summary. Few researchs compare the difference among different discrepancy.

Weakness:
1. I still think this paper is a small survey with respect to domain adaptation theory. But it seems that the aurthors have not claimed that.

2.In your abstract, you claim that "we propose new domain adaptation models and conduct experiments on real-world datasets that corroborate our theoretical findings". But I cannot find the new models you proposed (All models have been proposed before). So do you have over-claimed? Please explain the sentence  "we propose new domain adaptation models and conduct experiments on real-world datasets that corroborate our theoretical findings".

3. The theoreties discussed in your paper can only address the shift between marginal distributions, so when the conditional distributions are different, the thoreties cannot handle. The reason is that you omit the influence of feature transformation.

Please see the following papers to better understand:

1) How does the Combined Risk Affect the Performance of Unsupervised Domain Adaptation Approaches? AAAI 2021.

2) On Learning Invariant Representation for Domain Adaptatio. ICML 2019.

Above two papers both consider UDA with more deeply understanding.

4. Missing Reference:
A survey on domain adaptation theory.

---

> ### Author Response · Authors · 2022-10-25
> **RE: Review of Paper442 by Reviewer 7NW6**
>
> Thank you for your valuable comments on our work. Hopefully, our responses will help resolve your concerns.
>
> Answer 1: On one hand, in addition to the contributions to DA theory, this paper further answers the following questions: what are the pros and cons of the various divergence measures used in the DA literature (Sec 5)? which divergence would be the best for a specific dataset (Sec 6)? In this sense, we believe this work is more than a small survey w.r.t. DA theory.
>
> On the other hand, even in terms of DA theory, it's too arrogant for us to call our work a survey, especially compared with the thorough survey [3]. Our work looks like a survey in that we list the classical target error bounds and DA models in literature. The reason we lay out this way is that it's impossible for a theoretical paper to avoid these classical bound results. But we do not stop at just reviewing the existing results. We derive a more unified generalization bound that requires fewer assumptions than classical results. And we explicitly show the connections between the proposed framework with modern models. Thus, we have more innovations than a survey.
>
> Answer 2: The new models we propose (in the "Experiments" session) focus on aligning the domains with various divergence measures. The classical models use the divergence measures like Gaussian-kernel MMD (DAN), Proxy $\mathcal{A}$-distance(DANN), $L^1$-norm (MCD), disparity discrepancy (MDD), Wasserstein distance (WDGRL), etc. We propose new DA models which take divergence measures that are not covered before, including total variation, Neyman/Pearson $\chi^2$, KL, Reverse KL, SH, JS, and MMD with Laplacian kernels. In Table 4-6, we record the performance of the new models below the middle line of the table.
>
> Answer 3: See the response to Requested changes.
>
> Answer 4: Thank you for pointing out this important reference. We will add it to the manuscript.
>
> Response to Requested changes: We agree with the reviewer that there can be conditional shifts where the labeling functions of the source and target domains differ. In the manuscript, we only discuss the covariate shift, which is the common setting of DA literature, but do not cover the conditional shift in a more general sense. Thus we'd like to make the following revisions to the manuscript:
>
> The target error bound analysis that we propose still holds irrespective of the problem setting (covariate/conditional shift). The difference in solving these two problems lies in how to guide the algorithm design with the same error bound. In the covariate shift setting where it's assumed that the conditional distributions of $Y|X$ between the two domains are the same, the 3rd term $\lambda^*$ in Eq. (4) vanishes. Thus, it suffices to just learn the invariant representations while minimizing the source risk, as we have done in the experiments. In general settings where there can be conditional shifts, simply aligning the marginal distributions is not sufficient since $\lambda^*$ can't be neglected ([1], [2]). But we should also note that some researchers point out that $\lambda^*$ can be reduced to a rather small value if the hypothesis space is rich enough ([4]), making it still possible for the model to adapt well to the target domain by only minimizing the domain discrepancy and the source error even if there are conditional shifts.
>
> In terms of the algorithm design, as far as we know, there is no consensus reached on how to deal with the conditional shift in UDA. Since the key challenge takes root in the unavailability of labeled samples in the target domain, some works ([1], [5]) construct the pseudo labels for the target domain. However, the idea of constructing pseudo labels implicitly embeds the assumption that the labeling functions on the two domains should not be much different, given that the source domain knowledge is the only available reference for pseudo label construction. Moreover, although there is no consensus on the methodology, there are lessons that one should take in running the algorithm in case of conditional shifts. For example, the authors in [2] show that overtraining the feature transformation function and the discriminator may hurt generalization on the target domain. We take this lesson by adopting the early-stopping strategy in the experiments. Specifically, if the sum of the source error and the domain divergence, as a proxy signaling the overtraining, has not decreased for some epochs, we'll stop the network training. With this strategy, we can indeed reduce the impact of overtraining and obtain a pretty good model transferability as measured by the model performance on the target domain.
>
> [1] How does the Combined Risk Affect the Performance of Unsupervised Domain Adaptation Approaches?
>
> [2] On Learning Invariant Representation for Domain Adaptation.
>
> [3] A Survey on Domain Adaptation Theory.
>
> [4] Bridging Theory and Algorithm for Domain Adaptation.
>
> [5] Conditional Adversarial Domain Adaptation.

---

> > ### Comment · Reviewer_7NW6 · 2022-10-25
> > **To Response**
> >
> > I have read your response carefully.
> >
> > To be honest, it is really hard to say how large your contribution is.
> >
> > To response your answer 1, in machine learning, we pursue the generalisation of models. So what is the best divergence which can generalise well?  Additionally, I am very interested in how to estimate the TV distance. Do you think TV distance can be estimated by finite data? If it can be estimated by finite data, please explain why.
> >
> >
> > After revising, I think this paper can be accepted.

---

### Review · Reviewer_vrkj · 2022-10-21

**Summary Of Contributions:**

This paper studies the theory of using feature-level domain alignment to tackle the unsupervised domain adaptation problem. This paper first uncovers the long-standing theoretical gap between the widely used generalization bound analysis and various alignment-based domain adaptation methods. Then, the authors propose a unified generalization bound and use it to theoretically explain why several classical alignment-based adaptation methods perform well. To show the universality of the bound analysis, the authors provide a comprehensive comparison of common discrepancies and further derive new adaptation methods by using different divergences. Experimental results on three popular benchmarks show that the derived methods are competitive compared to classical models.

**Requested Changes:**

I would request the authors make the following important changes:

1. Add missing details of the newly derived methods and try to find why some promising divergences occasionally perform badly. Is this because different adaptation tasks require different hyperparameters or something else?

2. As for the presentation, I would first suggest not using "a unified domain adaptation framework" alone if the proposed bound is not universal for all domain adaptation methods. I would also suggest the authors give a detailed introduction to domain adaptation via feature alignment by a vivid figure and specific optimization process. Otherwise, readers who know little about domain adaptation may be curious about why domain alignments benefit target adaptation performance.

**Strengths And Weaknesses:**

### Strengths
1. This paper is well-motivated by uncovering the long-standing theoretical gap, comparing the previous generalization bound analysis, and proposing a new unified bound.
2. The proposed unified bound is universal for the alignment-based adaptation method, which is supported by theoretically explaining classical models.
3. Experiments empirically verify the effectiveness of the proposed bound by promising results of the derived new methods.
4. The extensive comparison of discrepancies is insightful for new alignment-based methods development in the future.

### Weaknesses
1. It seems this paper only focuses on feature alignment-based domain adaptation methods, supported by compared methods in Table 4 -Table 6. However, there is another mainstream line of methods based on self-training or pseudo-labeling. Is the proposed framework able to provide theoretical support for such methods? If not, for me, it is an overclaim to call the framework as a unified domain adaptation framework. It would be better to add feature-level domain alignment or similar words as the description.

2. I appreciate the theoretical contribution of this paper. But I have some questions as to the practical/empirical value of the proposed bound. As is observed by the authors, "there does not exist a typical distribution divergence measure that can universally outperform". I do not expect the authors to tackle this problem in this paper, but it would be better to provide an in-depth analysis of why this problem happens. In that way, follow-up works can benefit from this paper to develop more practical and universally effective adaptation methods.

3. Missing information on the newly derived methods in Table 4 - Table 6. What is the specific optimization objective for these methods? Do they share the same hyper-parameters including the loss coefficient?

---

> ### Author Response · Authors · 2022-11-04
> **Response to the weaknesses**
>
> Thank you for your insightful reviews. We respond to the weakness part in this section and defer the 'response to requested changes' to the next session.
>
> Answer 1: We agree with the reviewer that the models in the experiments focus on aligning the distributions of both domains to achieve good transferability. Also, we acknowledge that there is another line of research, as the reviewer reminds us, that applies self-supervised learning methods (including entropy minimization, pseudo-labeling, etc.) to unsupervised domain adaptation problems. However, the theoretical framework (the error bound analysis mainly) we propose still holds irrelevant to the methodologies/techniques used to solve the problem. The distribution-aligning domain adaptation approaches can indeed be derived directly from the theory and thus enjoy the theoretical guarantees, while semi-supervised learning methods are still in the empirical regime and can not be directly derived directly from the aforementioned error bound [1]. In the literature that focuses on self-training methods ([2,3]), the algorithm design is usually based on complicated assumptions. For example, to derive the theoretical bound, authors of [2] make a strong assumption that the model could perform rather well on the intermediate domains. Such strong assumptions make self-training methods pretty dependent on specific data format and hard to scale to different modalities [1].
>
> But still, we agree that it's inappropriate to call this work a unified domain adaptation framework, given that our analysis can only provide generalization guarantees for distribution-aligning approaches. We will be more modest in describing the contributions and will revise the manuscript accordingly.
>
> [1] Transferability in Deep Learning: A Survey.
>
> [2] Self-training Converts Weak Learners to Strong Learners in Mixture Models.
>
> [3] Theoretical Analysis of Self-Training with Deep Networks on Unlabeled Data.
>
> Answer 2: The practical value of the proposed bound lies in that it bridges the gap between the proliferating DA models that empirically take various divergence measures to align the domains and the needed theoretical support to verify the model's effectiveness. Although most DA models base themselves on the theoretical analysis in Ben-David et al. (2007) and Mansour et al. (2009), they do not take the divergence measures specified by the theories (namely, $\mathcal{H}$-divergence and discrepancy distance $dist_L^{\mathcal{H}}$). Moreover, some empirical works even lack the necessary theoretical explanation and only focus on practical performance. The bound we propose, together with the divergence analysis in Proposition 4-6, helps explain in more detail why some models can perform well theoretically. With this proposed framework, practitioners are given complete freedom in their choice of a specific divergence measure without worrying whether the corresponding model could theoretically work.
>
> The observation that "no single distribution divergence measure can universally outperform" is indeed the starting point of this research. In our work, we try to analyze this phenomenon by looking into the detailed structure of each divergence measure. Through the lens of computation costs, convergence rates and topologies, we provide the full profile of each divergence measure. As the work shows, each divergence measure has two sides. The calculation of f-divergence can be sensitive to the implementation of adversarial networks, while it enjoys fast convergence properties. Wasserstein distance can be limited in that it only aligns the first two moments of distributions. Evaluating MMD encounters the computational cost that grows quadratically with sample size, although it could measure higher-order moments of distributions. Ideally, it's better to give instructions on how to make trade-offs among these properties for a given dataset. We are still working on this direction, and hope the preliminary comparison over the multiple divergence measures could benefit the follow-up works.
>
> Answer 3: We will add the details of the newly proposed models to the manuscript, see the first part of 'Response to Requested changes'.

---

> ### Author Response · Authors · 2022-11-04
> **Response to the requested changes**
>
> Response to requested change 1: Thank you for your kind advice. We would make the following adjustments to the manuscript (due to some display issues with markdown, some mathematical expressions fail to get well displayed in this box. For more detailed and complete descriptions over the models, please see the revised manuscript in Section 6 'Experiments'):
>
> In this section, we enrich the DA models by proposing new model variants that take divergence measures which are not covered in literature before, including total variation (TV), Neyman $\chi^2$, Pearson $\chi^2$, KL, Reverse KL, SH, JS and MMD with Laplacian kernel functions.
>
> Let's first give a brief description over the proposed models. In the unsupervised DA setting, we are given $N_s$ labeled source samples with the $n$-th source sample denoted by $\{(x_n^s,y_n^s)\}$, together with $N_t$ unlabeled samples with the $n$-th target sample denoted by $\{x_n^t\}$. We'll first train a feature extractor $F(\cdot;\theta_f)$ parameterized by $\theta_f$ to extract the features. We denote the features extracted as $\mathcal{F}^s$ (for source features) and $\mathcal{F}^t$ (for target features). The feature extractor is followed by a discriminator network $D(\cdot;\theta_d)$ parameterized by $\theta_d$ to minimize the source error $\mathcal{L}(\{x_n^s,y_n^s\};\theta_f,\theta_d)$. Here, $\mathcal{L}$ takes cross-entropy loss for a classification problem and mean squared error for a regression problem.
> Meanwhile, good transferability requires the alignment of domains. Thus, the feature extractor is required to extract the domain-invariant features, guaranteed by minimizing the distance between the features $dist_{L}^{\mathcal{H}}(\mathcal{F}^s,\mathcal{F}^t)$. So the general objective for the proposed models is:
> $$
>     \mathop{\min}\limits_{\theta_f,\theta_d} \{\mathcal{L}(\{x_n^s,y_n^s\};\theta_f,\theta_d)
> +\lambda dist_L^{\mathcal{H}}(\mathcal{F}^s, \mathcal{F}^t) \},
> $$
> where $\lambda$ is a positive hyperparameter to be tuned. For illustration purposes, we sketch the general network flow for the proposed models in Figure 1. For the domain divergence measure $\textup{dist}_L^{\mathcal{H}}$ in the proposed models, we take specific divergence measures as mentioned in the last paragraph.
>
> For the reason why some promising divergence occasionally performs badly, we can provide some explanations with hindsight. In some cases, it can be attributed to the dataset. For example, one can find that the models with MMD are hardly comparable with the SOTA ones on Office-31 and Office-home datasets (mainly composed of object images), but achieve nearly the best performance on Digits dataset. The reason can be that MMD cannot take into account the geometry of the data distribution when estimating the discrepancy between two domains. Sometimes, the precision of measuring the distribution difference is prone to the implementation algorithms of the divergence measure, causing performance differences under different training schemes. Especially for $f$-divergence, minimizing the divergence is achieved by training an auxiliary adversarial network, the efficiency of which heavily depends on the complexity of the hypothesis space. Still, we should admit that prospective hindsight only works for the attributed analysis. Foresight is needed for designing brand-new DA algorithms, which is the research direction of our future work.
>
> Response to requested change 2: As the reviewer suggested, we would revise the manuscript accordingly. In addition, a graphic illustration for the model has been added to the manuscript, see Figure 1 in the revised manuscript.

---

### Review · Reviewer_PMEH · 2022-10-25

**Summary Of Contributions:**

To address the unsupervised domain adaptation problem, this paper investigates the theory of feature-level based domain alignment methods. This paper first points out that there is a theoretical gap between the widely used generalisation bound analysis and practical methods, and then further proposes a unified generalisation bound to theoretically explain the success of a variety of traditional alignment-based adaptation techniques. In addition, the authors also provide a conprehensive experimental comparisons new adaptation method and baselines. The results clearly demonstrate the proposed methods are competitive with classical models.

**Requested Changes:**

Please refer to the "Weaknesses" in the last section. I think all weakness should be addressed.

**Strengths And Weaknesses:**

Strengths:

1.The paper proposes a new unified bound that can theoretically explaining alignment-based adaptation method.

2.Based on the proposed bound, the paper further proposes new methods based on different distance estimator. The experimental results demonstrate the effectiveness of the proposed bound.

Weaknesses:

1. There are so few details in the proposed methods in this paper.  For example, how can we optimize these methods? I think that the authors should mention more details about the optimizatio process.

2. The paper claims that the bound can be generalized for both classification and regression tasks, but there are regression experimental comparisons in this paper.

3. The paper only includes a part of baseslines in the Table 4-6, but neglect more recent baselines, such as CAN and f-DAL mentioned in Table 2.

---

> ### Author Response · Authors · 2022-11-04
> **RE: Review of Paper442 by Reviewer PMEH**
>
> Response to Weakness 1: Thank you for your kind advice. We would make the following adjustments to the manuscript (due to some display issues with markdown, some mathematical expressions fail to get well displayed in this box. For more detailed and complete descriptions over the models, please see the revised manuscript in Section 6 'Experiments'):
>
> Let's first give a brief description over the proposed models. In the unsupervised DA setting, we are given $N_s$ labeled source samples with the $n$-th source sample denoted by $\{(x_n^s,y_n^s)\}$, together with $N_t$ unlabeled samples with the $n$-th target sample denoted by $\{x_n^t\}$. We'll first train a feature extractor $F(\cdot;\theta_f)$ parameterized by $\theta_f$ to extract the features. We denote the features extracted as $\mathcal{F}^s$ (for source features) and $\mathcal{F}^t$ (for target features). The feature extractor is followed by a discriminator network $D(\cdot;\theta_d)$ parameterized by $\theta_d$ to minimize the source error $\mathcal{L}(\{x_n^s,y_n^s\};\theta_f,\theta_d)$. Here, $\mathcal{L}$ takes cross-entropy loss for a classification problem and mean squared error for a regression problem.
> Meanwhile, good transferability requires the alignment of domains. Thus, the feature extractor is required to extract the domain-invariant features, guaranteed by minimizing the distance between the features $dist_{L}^{\mathcal{H}}(\mathcal{F}^s,\mathcal{F}^t)$. So the general objective for the proposed models is:
> $$
>     \mathop{\min}\limits_{\theta_f,\theta_d} \{\mathcal{L}(\{x_n^s,y_n^s\};\theta_f,\theta_d)
> +\lambda dist_L^{\mathcal{H}}(\mathcal{F}^s, \mathcal{F}^t) \},
> $$
> where $\lambda$ is a positive hyperparameter to be tuned. For illustration purposes, we sketch the general network flow for the proposed models in Figure 1. For the domain divergence measure $\textup{dist}_L^{\mathcal{H}}$ in the proposed models, we take divergence measures which are not covered in literature before, including total variation (TV), Neyman $\chi^2$, Pearson $\chi^2$, KL, Reverse KL, SH, JS and MMD with Laplacian kernel functions.
>
> Response to weakness 2: As you suggested, we will design the experiments on regression tasks and add the analysis to the revised manuscript. Typically, we test the efficacy of the new DA model variants on two image regression datasets: MPI3D and dSprites. MPI3D is a simulation-to-real dataset of 3D objects, consisting of three domains: Toy (T), RealistiC (RC) and ReaL (RL). Each domain contains 1,036,800 images, with every image having two continuous factors of variations to be predicted: a rotation about a vertical axis at the base and a second rotation about a horizontal axis. We evaluate all methods on six transfer tasks:RL$\to$RC, RL$\to$T, RC$\to$T, RC$\to$RL, T$\to$RL, and T$\to$RC. For each task, we separately predict the two rotation angles and record the sum of mean absolute error (MAE) over the two predictions as the transfer performance of this task. dSprites is a 2D synthetic dataset composed of three domains: Color (C), Noisy (N) and Scream (S). Each domain consists of 737,280 images with three variants of continuous factors to be predicted: scale, position X, and position Y. We evaluate all methods on 6 transfer tasks: C$\to$N, C$\to$S, N$\to$C, N$\to$S, S$\to$C, S$\to$N. Similar to the experiment on the previous dataset, we record the sum of MAE over the three predictions in each transfer task as the transfer performance. All experiments in this section are run with the same device as in the previous section.
>
> We compare with the following DA models that can be applied to regression problems: DAN, DANN, MCD, AFN and JDOT. For the MPI3D and dSprites experiments, ResNet-18 pretrained on ImageNet is adopted as the backbone feature extractor. The regressors are a 2-layer convolutional neural network with BatchNorm and ReLU as the activation function, followed by an average pooling layer and a 1-layer neural network with width 1024. For models using $f-$divergences, we introduce a domain critic to estimate the $f-$divergence in a min-max manner. The domain critic is a 2-layer convolutional neural network followed by a 1-layer neural network with ReLU as activation functions. We solve the min-max problem using gradient reversal layers. We take the mini-batch SGD optimizer with the Nestorov momentum 0.95 and the learning rates of the regressors are set to be 10 times that of the feature extractor. We follow the standard experimental protocol for unsupervised DA and report the MAE on the two datasets in Table 7 and 8 in the revised manuscript.
>
> For more details, please see Section 6.2 in the revised manuscript.
>
> Response to weakness 3: Thank you for reminding us. We have added the experimental results of CAN and f-DAL to the manuscript. See Table 4-6 in the revised manuscript.

---

### Author Response · Authors · 2022-11-04
**Summary of revisions**

We thank all reviewers for their helpful feedback, it has helped us strengthen the paper.

We are glad that the reviewers found the problem that we address well-motivated [vrkj], and the writing solid [TNW6]. Reviewers found the comparison of multiple divergence measures extensive and detailed [vrkj, TNW6] and the empirical analysis informative and insightful [PMEH, vrkj, TNW6].

There were some concerns, which we address in our responses to individual reviewers. We also incorporated the feedback and uploaded the new version of our paper (with the modifications made in the colored background):

- We better explain the UDA theory when there are conditional shifts [TNW6].
- We clarify that the framework we propose is unified in that it provides more general theoretical guarantees for the distribution-aligning DA methods [vrkj].
- We add more experimental details to the revised manuscript, including the training objectives [vrkj, PMEH] and a graphical illustration of the distribution-aligning methodologies [vrkj].
- We design additional regression transfer tasks to verify the proposed framework's effectiveness and add the performance of missing models [PMEH].

Please see individual responses below for other specific changes and more details.

---

### Public Comment · ~Matteo_Pagliardini1 · 2022-11-23
**Clarification on section 4.1.1**

Dear authors,

I thank you for your work and salute the much-needed effort in addressing the gap between known bounds and existing DA methods. I have one question regarding section 4.1.1, I was wondering if you could detail your derivation of equation 5 which represents the training objective of Saito et al. (2018). From the original paper, given $h_1=f_1 \circ g$ and $h_2=f_2 \circ g$, two predictors with shared encoder $g$, the objective is described as a three-step mechanism (with slightly altered notations: $L\equiv\mathcal{L}$, $\ell\equiv L$):

* Step 1: Train $h_1$ and $h_2$ by minimizing the risk over $P$: $\min_{h_1,h_2 \in H} L_P(h_1,h_s) + L_P(h_2,h_s)$
* Step 2: Freeze $g$, and maximize discrepancy over $Q$: $\min_{f_1,f_2} L_P(h_1,h_s) + L_P(h_2,h_s) - \mathbb{E}_{x\sim Q} \ell(f_1(g(x)), f_2(g(x)))$
* Step 3: Freeze $f_1$ and $f_2$ and minimize discrepancy over $Q$: $\min_{g} \mathbb{E}_{x\sim Q} \ell(f_1(g(x)), f_2(g(x)))$

From those steps, how do you reach equation 5?

Maybe another minor suggestion would be to bring the short proof of theorem 3 in the main paper, as well as add more comments to contrast it with the steps taken in Ben David et al. (2007).

Thank you for your time,

---

> ### Author Response · Authors · 2022-11-27
> **Re: Clarification of Equation (5) in Section 4.1.1**
>
> Hi Matteo,
> We really appreciate you for looking into the details of our work. There are actually some abuses of notations in aligning the terminologies between our work and Saito et al. (2018). We would make the following adjustments to the manuscript. A better formulation of Eq. (5) should be:
>
> \begin{equation*}
>     \mathcal{L}\_P(h\_1,h\_s) + \mathcal{L}\_P(h\_2,h\_s) - \mathbb{E}\_{x \sim Q}L(h\_1(x),h\_2(x)),
> \end{equation*}
>
> where $h_1$ and $h_2$ can be decomposed as $h_1 = f_1 \circ g$, $h_2 = f_2 \circ g$, and $L(\mathbf{u},\mathbf{v})=\frac{1}{m} \sum_{i=1}^m \vert u_i - v_i\vert$ with $\mathbf{u},\mathbf{v}\in \mathbb{R}^m$. The training of model parameters by Saito et al. (2018) is conducted in three steps alternatively:
>
> (1.) step $\textbf{A}$: $(\hat{g},\hat{f\_1}, \hat{f\_2}) = \textup{argmin}\_g \textup{argmin}\_{f\_1,f\_2} \mathcal{L}\_P(f\_1 \circ g,h\_s) + \mathcal{L}\_P(f\_2 \circ g,h\_s)$;
>
> (2.) step $\textbf{B}$: $(f\_1^*, f\_2^*) = \textup{argmin}\_{f\_1,f\_2} \mathcal{L}\_P(f\_1 \circ \hat{g},h\_s) + \mathcal{L}\_P(f\_2 \circ \hat{g},h\_s) - \mathbb{E}\_{x \sim Q}L( f\_1 \circ \hat{g}(x), f\_2 \circ \hat{g}(x) )$;
>
> (3.) step $\textbf{C}$: $g^* = \textup{argmin}\_{g} \mathbb{E}\_{x \sim Q}L( f\_1^* \circ g(x), f\_2^* \circ g(x) )$.
>
>
> The core message we'd like to convey is that the optimization steps actually minimize the source error with the domain discrepancy measured by total variation distance. Typically, given the optimal feature extractor $\hat{g}$ trained to achieve the minimal source error, step B trains two classifiers $f_1$ and $f_2$ to further minimize the source error and \textit{measure} the domain discrepancy in an adversarial way. We will later prove that  $\min -\mathbb{E}_{x \sim Q} L(f_1^* \circ \hat{g}(x),f_2^* \circ \hat{g}(x))$ returns the domain divergence $2 D\_{TV}(P,Q)$, given that $f_1^* \circ \hat{g}$ and $f_2^* \circ \hat{g}$ can simultaneously perform well on the source domain. Then, step $\textbf{C}$ just further minimizes the domain divergence.
>
> Next, we show that step B trains two classifiers $f_1$ and $f_2$ to further minimize the source error and \textit{measure} the domain discrepancy. Expressing the total variation distance in the form of discrepancy distance, we have
>
>
> \begin{equation*}
> D\_{\textup{TV}}(P,Q) = \frac{1}{2} \mathop{\textup{sup}} \limits\_{h\_1,h\_2} \big\vert \mathbb{E}\_{x \sim P}L(h\_1(x),h\_2(x) - \mathbb{E}\_{x \sim Q}L(h\_1(x),h\_2(x))\big\vert.
> \end{equation*}
>
> By assuming that the classifiers $h_1,h_2$ will work well on the source domain, the authors suggest that the term $\mathbb{E}\_{x \sim P}L(h\_1(x), h\_2(x))$ is small enough to omit. Thus the following equivalence holds:
> \begin{equation*}
> \begin{aligned}
> & \mathop{\textup{min}} \limits\_{h\_1, h\_2}  \mathcal{L}\_P(h\_1,h\_s) + \mathcal{L}\_P(h\_2,h\_s) - \mathbb{E}\_{x \sim Q}L(h\_1(x),h\_2(x))\\
> \Leftrightarrow & \mathop{\textup{min}} \limits\_{h\_1, h\_2}\mathcal{L}\_P(h\_1,h\_s) + \mathcal{L}\_P(h\_2,h\_s) + \mathop{\textup{sup}} \limits\_{h\_1,h\_2} \big\vert \mathbb{E}\_{x \sim Q}L(h\_1(x),h\_2(x)) \big\vert \\
> \Leftrightarrow & \mathop{\textup{min}} \limits\_{h\_1, h\_2}\mathcal{L}\_P(h\_1,h\_s) + \mathcal{L}\_P(h\_2,h\_s) + \mathop{\textup{sup}} \limits\_{h\_1,h\_2} \big\vert \mathbb{E}\_{x \sim P}L(h\_1(x),h\_2(x)) - \mathbb{E}\_{x \sim Q}L(h\_1(x),h\_2(x))\big\vert\\
> \Leftrightarrow & \mathop{\textup{min}} \limits\_{h\_1, h\_2} \mathcal{L}\_P(h\_1,h\_s) + \mathcal{L}\_P(h\_2,h\_s) + 2 D\_\textup{TV}(P,Q)\\
> \end{aligned}
> \end{equation*}

---

### Decision · Action_Editors · 2022-11-24

**Recommendation:** Accept as is

**Comment:**

This paper rethinks the current theory and methods for domain adaptation and identify a gap between theory and practice. To fill this gap, this paper proposes a unified generalization bound which can explain the success of various domain adaptation methods. The proposed theory is also validated via comprehensive results.

After several revisions, all the reviewers think the paper makes a good contribution and the presentation is of good quality now. Based on the review comments and suggestions, I would like to recommend acceptance of this paper.

**Audience:**

yes

**Claims And Evidence:**

yes